# High-throughput fitness screening and transcriptomics identify a role for a type IV secretion system in the pathogenesis of Crohn's disease-associated *Escherichia coli*

Wael Elhenawy[1,2], Sarah Hordienko[1,7], Steven Gould[1,7], Alexander M. Oberc[1], Caressa N. Tsai[1], Troy P. Hubbard[3,4], Matthew K. Waldor [3,4,5] & Brian K. Coombes [1,2,6 ✉]

Adherent-invasive *Escherichia coli* (AIEC) are pathogenic bacteria frequently isolated from patients who have Crohn's disease (CD). Despite the phenotypic differences between AIEC and commensal *E. coli*, comparative genomic approaches have been unable to differentiate these two groups, making the identification of key virulence factors a challenge. Here, we conduct a high-resolution, in vivo genetic screen to map AIEC genes required for intestinal colonization of mice. In addition, we use in vivo RNA-sequencing to define the host-associated AIEC transcriptome. We identify diverse metabolic pathways required for efficient gut colonization by AIEC and show that a type IV secretion system (T4SS) is required to form biofilms on the surface of epithelial cells, thereby promoting AIEC persistence in the gut. *E. coli* isolated from CD patients are enriched for a T4SS, suggesting a possible connection to disease activity. Our findings establish the T4SS as a principal AIEC colonization factor and highlight the use of genome-wide screens in decoding the infection biology of CD-associated bacteria that otherwise lack a defined genetic signature.

[1] Department of Biochemistry and Biomedical Sciences, McMaster University, Hamilton, ON, Canada. [2] Michael G. DeGroote Institute for Infectious Disease Research, Hamilton, ON, Canada. [3] Division of Infectious Diseases, Brigham & Women's Hospital, Boston, MA, USA. [4] Department of Microbiology, Harvard Medical School, Boston, MA, USA. [5] Howard Hughes Medical Institute, Boston, MA, USA. [6] Farncombe Family Digestive Health Research Institute, Hamilton, ON, Canada. [7] These authors contributed equally: Sarah Hordienko, Steven Gould. ✉email: coombes@mcmaster.ca

The finely tuned interactions between the microbiome and the immune system are fundamental to gut homeostasis. Perturbing this equilibrium can have detrimental consequences on human health, including the development of inflammatory bowel diseases like Crohn's disease (CD). Genome-wide association studies have identified multiple human gene variants that increase the risk of developing CD[1,2]; however, the genetic make-up of the host alone does not fully account for the dysregulated immune response to the microbiome in CD. There is widespread consensus that non-genetic factors—including environmental, xenobiotic, and microbial—play a significant role in CD development in genetically susceptible individuals[3]. Animal models of CD have provided consistent evidence supporting the role of bacteria in driving gut inflammation. For example, animal models with genetic predispositions to colitis do not develop disease under germ-free conditions, indicating that the microbiome is an active disease modifier[4,5]. Several studies have employed metagenomics to profile the CD-associated gut microbiome in an attempt to identify potential causative organisms[6,7]. While a definitive microbial signature of CD has not been precisely defined, the expansion of *Escherichia coli* in the gut is frequently reported in CD-associated dysbiosis, particularly at the mucosal surface. A specific *E. coli* pathovar, known as adherent-invasive *E. coli* (AIEC), is associated with CD and has in vivo traits that distinguish it from commensal *E. coli* at the phenotypic level[8,9]. For example, AIEC invade into intestinal epithelial cells[10], outcompete commensals in the inflamed gut through enhanced use of oxidized metabolites, and evade inflammatory host defenses[11–14]. Despite substantial evidence supporting AIEC as a disease modifier in CD, the mechanisms used by AIEC to colonize the intestine remain largely unknown.

We previously used experimental evolution to track the AIEC fitness landscape in vivo, revealing that hypermotility and enhanced consumption of short chain fatty acids are selective traits for AIEC outgrowth in the gut[9]. While this approach helped uncover the selective adaptations acquired by AIEC over chronic timescales in the host, this method lacked the resolution necessary for a comprehensive analysis of within-host gene essentiality and dispensability. Transposon-insertion sequencing (TIS) has been widely used to interrogate bacterial genes required for survival in different environments. In TIS experiments, transposon-mediated mutagenesis is used to generate a high-density library of mutants, which is then exposed to different selective pressures. Using highly parallel sequencing, the fluctuations in mutant abundance under selection can be monitored with high resolution, where underrepresented mutants represent the genetic loci that give rise to a selective advantage[15,16].

Here, we used TIS to uncover the genetic inventory that enables AIEC to proliferate in the host environment. The TIS-based screen identified 377 genetic loci required by AIEC to efficiently colonize the murine gut that were otherwise dispensable for growth in vitro. In parallel, we performed comparative transcriptomics to identify the AIEC gene expression profile that was specific to the host environment. This approach identified diverse metabolic pathways required for efficient gut colonization by AIEC and identified a key role in virulence for a type IV secretion system (T4SS), a nanomachine that mediates several functions including protein secretion, DNA transfer, and biofilm formation[17]. We show that AIEC uses type IV secretion to form biofilms on epithelial cells in vivo and to survive in the host gut, thereby enabling a bacterial lifestyle that is known to be intrinsically more resistant to antibiotics and host immunity. Finally, we found that the T4SS is enriched among *E. coli* strains obtained from CD patients compared to healthy individuals. Together, these results further our understanding of AIEC pathogenesis and unravel new mechanisms employed by AIEC to thrive in the CD environment.

## Results

**High-resolution genomic screening for AIEC fitness determinants in vivo.** A comprehensive genome-scale screening strategy was established to identify within-host fitness determinants of AIEC involved in colonizing the murine gut (Fig. 1a). The mariner transposon we used integrates at T-A sites in the genome, enabling the generation of highly saturated mutant libraries at sub-genic resolution[18]. Our initial attempts to create a complex transposon mutant library in AIEC were hampered by jackpot insertion sites and retention of the transposon vector, resulting in a mutant library with low saturation (Supplementary Fig. 1). To circumvent these issues, we modified the original vector carrying the mariner transposon and its cognate transposase[19] by replacing the transposase promoter with an inducible promoter and adding a counter-selectable marker to avoid retention of the plasmid backbone (Supplementary Fig. 1). Using this new custom transposon system, we generated a second AIEC library of ~600,000 insertion mutants that collectively saturated ~65% of all T-A dinucleotide sites in the genome (Fig. 1b).

To identify AIEC genes that promote fitness in a CD model[20], three C57BL/6N mice were orally inoculated with the high-density AIEC mutant library. After 3 days, the mutant population was retrieved from the cecum and the abundance of each mutant relative to the input inoculum was determined using deep sequencing and the ConARTIST analytic pipeline[18]. This method uses simulation-based normalization to overcome the effects of non-selective stochastic processes including infection bottlenecks and sampling errors, facilitating identification of mutant genotypes that are depleted due to selection rather than stochastic loss[18]. In line with this, significant correlation between the different in vivo mutant libraries was observed (Supplementary Fig. 2). Using this approach, we identified 377 genes that enabled AIEC colonization of the host, yet were dispensable for growth in vitro (Fig. 1c). We previously showed that swimming motility is important for AIEC to invade into the intestinal mucosa and establish gut colonization[9]. Indeed, the TIS data showed that mutants lacking flagellin (*fliC*), and the flagellum-specific ATP synthase (*fliI*) were significantly depleted in vivo (Supplementary Data 1). Furthermore, mutants lacking the ability to make cell wall glycolipids were detected at low abundance in vivo, in agreement with their role in the evasion of the immune system, thereby validating our genomic approach (Fig. 1c). This included *wecA*, an undecaprenyl-phosphate alpha-*N*-acetylglucosaminyl 1-phosphate transferase required for the synthesis of lipopolysaccharides, and the enterobacterial common antigen polymerase, *wzyE*[21,22].

To begin to investigate the cellular pathways that contribute to AIEC fitness in the host, in vivo-depleted loci were assigned to clusters of orthologous groups (COGs) (Fig. 1d and Supplementary Data 1)[23]. The COG analysis indicated that ~50% of the AIEC in vivo fitness genes were involved in metabolism, with 18% of the total loci involved in carbohydrate metabolism and nutrient transport. Interestingly, mutants in four genes involved in arginine biosynthesis, *argBCGH*, were significantly depleted in vivo, indicating an important role for this pathway in the gut[24]. In accord with this finding, additional in vivo fitness genes included a carbamate kinase (NRG857_19070) involved in the synthesis of arginine and pyrimidines[25]. Several studies have shown that inflammation reshapes the metabolic potential of the gut, facilitating the expansion of certain members of Enterobacteriaceae[14,26,27]. Indeed, AIEC have acquired several adaptations to compete in the inflamed gut environment[28]. For

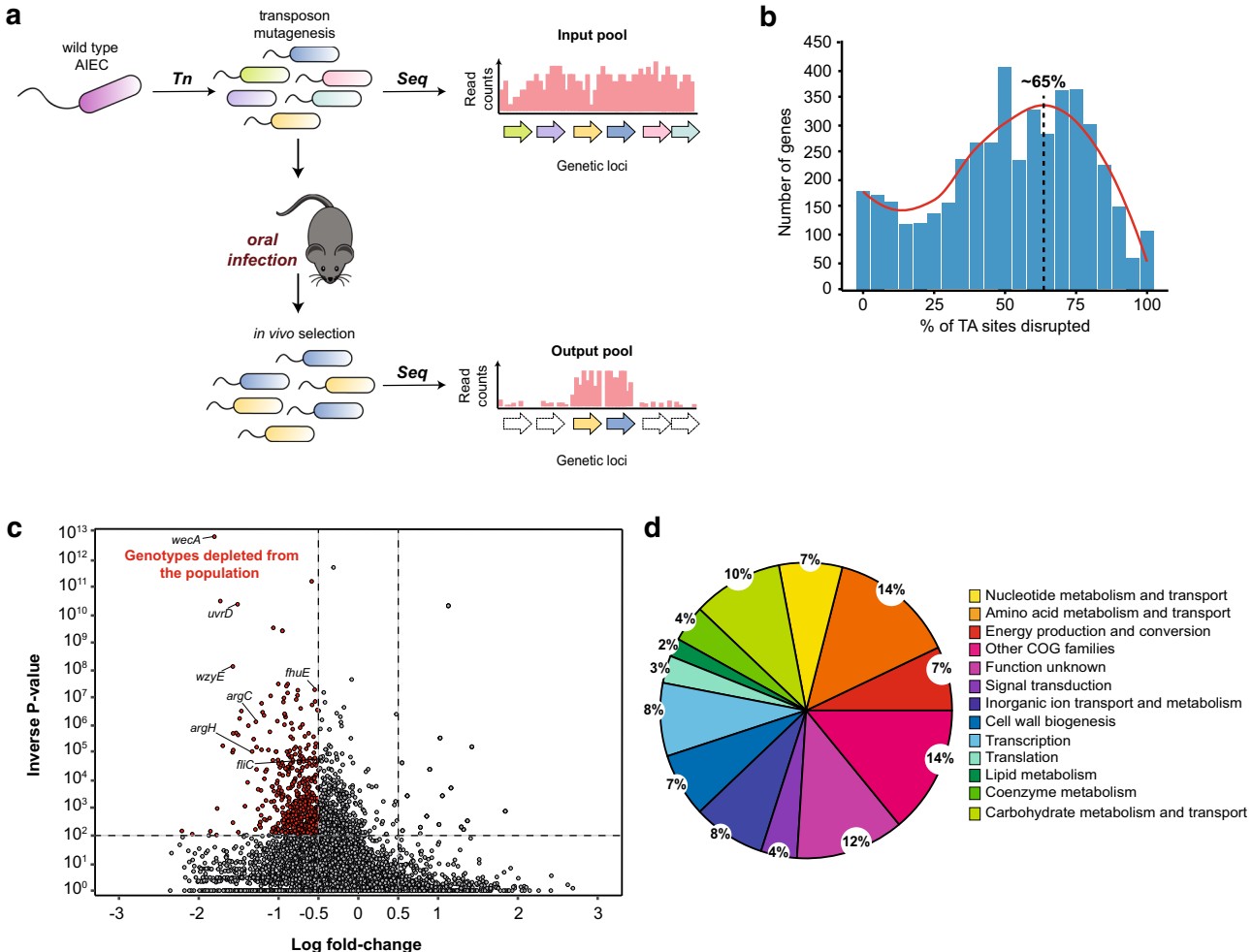

**Fig. 1 Identification of the genetic requirements for AIEC fitness in the gut. a** Experimental scheme for the transposon (Tn) sequencing (Seq)-based screen for AIEC colonization determinants in the host environment. **b** Histogram showing the percentage of TA sites disrupted per gene in AIEC str. NRG857c input library using the modified transposon system. **c** Volcano plot showing the mean fold change in genotype abundance of three genomic libraries versus the statistical significance (inverse *P* value); *n* = 3 biologically independent animals infected with the mutant libraries. Computation of *P* values is constitutive of the ConARTIST analysis pipeline using Mann–Whitney *U* test. Red dots represent the genotypes with significant variation in abundance in vivo (*P* value <0.01 in all three biological replicates). **d** Functional assignment of the AIEC in vivo-depleted loci to COG families.

example, upregulation of nitric oxide synthase (*Nos2*) in the gut during inflammation elevates nitrate levels that can be used by *E. coli* as a terminal electron acceptor to outcompete other commensal bacteria during anaerobic growth[27,29]. Our data showed that insertions in the AIEC nitrate reductase, *narH* (NRG857_07280), had lower fitness in the gut (Supplementary Data 1). Reactive nitrogen species produced by Nos2 can also generate galactarate from galactose oxidation that can expand the metabolic niche for gut pathogens[14,26]. Consistent with this model, our TIS data showed that inactivation of the galactarate permease, *garP*, significantly reduced AIEC fitness in vivo. Together, these results pointed to nitrate metabolism as having an important role in shaping the metabolic fitness of AIEC in the gut. *Escherichia coli* has enzymes that can detoxify nitric oxide to the less toxic nitrous oxide using the NorVW reductases, regulated by the transcriptional activator, NorR[30]. We found that insertion mutations in all three members of the *norRVW* operon reduced AIEC in vivo fitness, suggesting that detoxifying reactive nitrogen species provides a selective advantage to AIEC in the gut. Many loci involved in amino acid metabolism and transport were important for AIEC growth in vivo as well

(Fig. 1d). This finding is consistent with recent data on the role of amino acid metabolism in maximizing AIEC fitness[12]. The genes we identified as being important for AIEC intestinal colonization were dispensable for AIEC growth in vitro, thereby revealing the inventory of genes vital to AIEC growth within a host.

We detected several two-component system mutants among the in vivo-depleted AIEC loci, including PhoQ and EnvZ (Supplementary Data 1) inferring the importance of regulatory mechanisms governing the transition to the host environment. Out of ~30 two-component systems in *E. coli*, PhoPQ and OmpR/EnvZ are known to control major regulons needed for bacterial adaptation to the host environment[31]. Our analysis revealed that *phoP, phoQ, envZ*, genes were all required for AIEC in vivo fitness (Supplementary Data 1). Our data confirmed the role of quorum-sensing in AIEC virulence[32] whereby mutants with insertions in the genes encoding for the quorum-sensing histidine kinase, QseC, and its response regulator, QseB, were depleted in vivo. In addition to identifying genes with known or inferred functions, 12% of the AIEC in vivo-depleted loci encode for hypothetical proteins, suggesting that many mechanisms mediating AIEC pathogenesis during CD remain to be characterized.

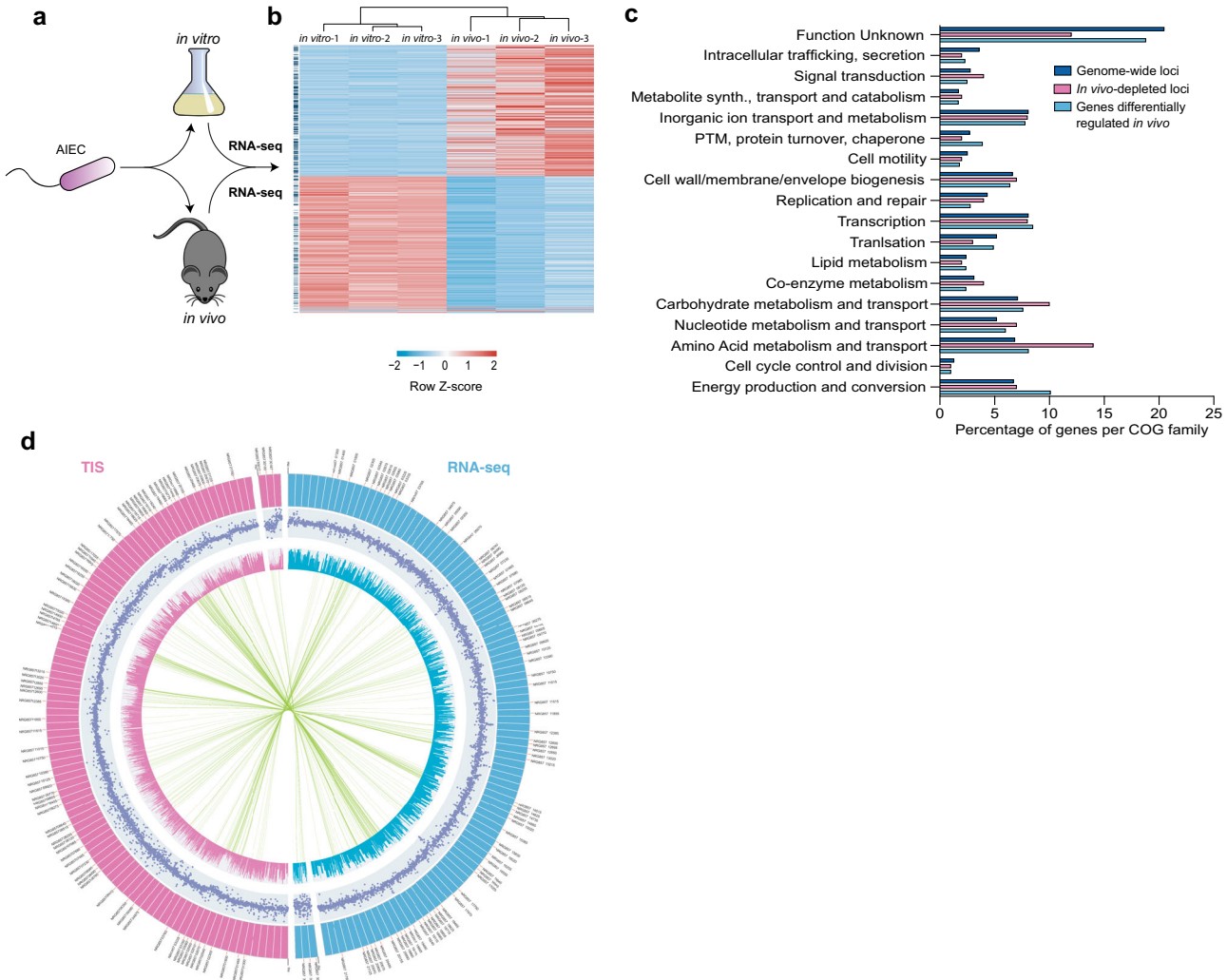

**Fig. 2 Combined genome-scale approaches to investigate AIEC infection biology. a** Identification of the AIEC intestinal gene expression profile using comparative transcriptomics. **b** Heatmap with hierarchical clustering showing the relative gene expression of different AIEC loci in vivo and in vitro. Only statistically significant transcripts per million (TPM) values are shown (FDR-adjusted P value <0.05). Individual rows marked with a blue line (to left of panel) represent genes involved in metabolic processes. **c** Bar graph representing the percentage distribution of differentially regulated genes and in vivo-depleted loci relative to the whole genome across different COG families. Hits from TIS and RNA-sequencing shown were obtained from $n = 3$ biologically independent animals. **d** Comparisons of the AIEC in vivo fitness genes and transcriptomic profiles. Mean fold change of mutant abundance (pink track) and differential gene expression (blue track) in vivo are displayed on the log scale (dot plot) and the linear scale (histogram). Lines are connecting the AIEC loci that were common between the two datasets.

**Transcriptional reprogramming in the host environment is central to AIEC metabolic adaptability.** Our TIS screen revealed the presence of several two-component systems that were required for AIEC fitness in vivo, suggesting that AIEC undergo regulatory adaptation to the host environment. To assess this and to complement the TIS screen, we performed in vivo RNA-sequencing to measure global AIEC gene expression in the host setting. Three days after AIEC infection, total RNA was isolated from the mouse cecum and sequenced to characterize AIEC gene expression (Fig. 2a). Commensal *E. coli* constitutes a small percentage of the mouse microbiome[33], and segregating commensal and AIEC-specific transcripts can be challenging. However, the streptomycin treatment in our AIEC infection model markedly depletes commensal *E. coli* from the feces, which was confirmed by plating on coliform-selective agar. The Illumina HiSeq platform was used to ensure that we had sufficient read coverage of the AIEC in vivo transcriptome. After filtering out host sequences, ~1 million reads mapped to the reference genome of AIEC strain NRG857c with mean read depth of ~16 reads per base. To

identify the host-specific transcriptome of AIEC, we also performed RNA-seq on AIEC grown in vitro for approximately the same number of generations, and computed the differential expression of AIEC genes in vitro and in vivo using DeSeq2 (ref. [34]) (Fig. 2a and Supplementary Data 1).

Hierarchical clustering confirmed that the biological replicates of each condition grouped together (Fig. 2b). Additionally, there was significant correlation between the different in vivo transcriptomes, demonstrating reproducibility within our results (Supplementary Fig. 3). We identified 1700 genes that were differentially regulated in the host environment, with 851 genes upregulated in vivo (FDR-adjusted P value <0.05) (Fig. 2b). Among these, 1634 genes could be assigned to 17 COGs, with nearly half (46%) belonging to metabolic pathways, consistent with the TIS data that revealed the importance of metabolic adaptability (Supplementary Data 1). There were 324 genes of unknown function that could not be assigned to a specific COG, resulting in the distribution of differentially regulated AIEC genes across the different COG families being similar to that of the

in vivo fitness genes (Fig. 2c). Among the in vivo fitness and differentially regulated gene datasets, certain COG families appeared enriched as compared to their relative distribution at the whole-genome level. For example, 10% of the differentially regulated loci were involved in energy production and conversion which is higher than the representation of this COG family in the AIEC genome (6.7%). Similarly, 24% of the in vivo fitness genes are involved in carbohydrate and amino acid metabolism compared to an abundance of 14% of these COG families in the genome (Fig. 2c). Included among the upregulated metabolic genes was the *pdu* operon, which is required by AIEC to use the fucose derivative, propanediol (Supplementary Data 1)[35], and was previously shown to be important for AIEC survival in macrophages and to promote inflammation[35,36]. We also detected the upregulation of *nagA* and *nagB*, which are required for the conversion of *N*-acetyl-glucosamine to glucose-6-phosphate[37]. *N*-acetyl-glucosamine is liberated through the degradation of host mucus by the gut microbiome, which makes it one of the most abundant sugars in vivo[38]. In addition to being a carbon source for *E. coli*, *N*-acetyl-glucosamine is essential for cell wall synthesis[39]. The gene encoding the *N*-acetyl-glucosamine transporter, *nagE*[40,41], was markedly upregulated in vivo, suggesting that mucus-derived *N*-acetyl-glucosamine is a prominent energy source and cell wall building block for AIEC in the gut environment. Additionally, two members of the galactitol phosphotransferase uptake system were upregulated in vivo. Galactitol is a sugar alcohol in the gut that can be used by *E. coli* as a carbon source. The galactitol utilization operon has been previously identified as a target of *E. coli* adaptation to the gut environment, implying its importance for intestinal colonization[9,42]. We also detected the upregulation of several genes involved in lipid metabolism. This included the *eut* operon, which is required for the use of ethanolamine[43], a lipid head group molecule that is abundant in vivo and whose acyl derivatives are consumed by CD-associated bacteria[44]. Many metabolic enzymes require iron as a co-factor, making iron acquisition an activity critical to bacterial colonization of the gut[45]. Siderophores are soluble molecules secreted by bacteria that sequester iron with high affinity and facilitate iron acquisition by bacteria[45]. Our data revealed the upregulation of several genes involved in iron uptake, including the siderophore membrane transporters, *fepD* and *fecA*, involved in the uptake of enterobactin and ferrichrome, respectively[46,47]. As a defense mechanism, the host can produce the siderophore-binding lipocalin-2, which sequesters enterobactin. However, some bacteria secrete stealth siderophores, like yersiniabactin and salmochelin, that escape lipocalin-2 sequestration[45,48]. There was significant upregulation of yersiniabactin synthesis genes by AIEC in vivo, suggesting that AIEC uses this stealth siderophore to overcome nutritional immunity in the gut (Supplementary Data 1). Together, these data highlight the importance of metabolic flexibility and metal acquisition for AIEC to adapt to the gut environment.

**TIS and in vivo RNA sequencing identify a T4SS as a major AIEC colonization factor**. We compared the list of genes that were upregulated specifically in the gut to the set of genes that were required for in vivo growth identified by TIS as a means to prioritize new infection biology. We found that several genes were required for robust gut colonization and also upregulated upon entering the intestinal environment, suggesting critical functions in AIEC pathogenesis and colonization (Fig. 2d and Supplementary Data 1). This included arginine biosynthesis and the ferrichrome membrane receptor, *fecA*, confirming that amino acid metabolism and iron acquisition are critical factors to AIEC

colonization. This list also included three genes in the *frdABCD* operon that encode the different subunits of fumarate reductase required for using fumarate as an electron acceptor by *E. coli* during anaerobic respiration[49]. Our data revealed an additional eight genes with unknown function that were both upregulated in the host and required for fitness, suggesting the likelihood of other uncharacterized AIEC virulence mechanisms for future prioritization.

While the majority of the AIEC fitness components identified in our screen belonged to metabolic pathways, our data showed that multiple genes coding for a T4SS were critical to AIEC colonization of the host (Figs. 2d, 3a, and Supplementary Data 1). T4SSs are multifunctional nanomachines that span the cell envelopes of Gram-negative and Gram-positive bacteria to produce surface pili. In addition to DNA exchange, T4SSs support protein secretion and biofilm formation[50,51]. In *E. coli*, the T4SS is encoded by the *tra* operon, a homolog of the prototypical VirB/VirD4 system in *Agrobacterium tumefaciens*[50]. TraG is an inner membrane protein that is important for pilus assembly, while TraBH is an essential component of the core T4SS complex[52,53]. Both *traG* and *traBH* were required for within-host fitness of AIEC and were upregulated in the gut. Additional T4SS components that were either upregulated or essential for colonization included the main regulator, TraY, and the major pilin, TraA (Fig. 3a).

**T4SS mediates biofilm formation by AIEC on epithelial cells**. To investigate the role of T4SS in AIEC pathogenesis, we created a non-functional T4SS mutant by deleting the major pilin gene, *traA*[54]. To confirm that type IV secretion was required for AIEC fitness in vivo, we competed the Δ*traA* mutant with the isogenic wild-type strain in a CD infection model. The Δ*traA* mutant was outcompeted by the wild-type AIEC strain in all gut tissues, corroborating the TIS data (Fig. 3b). Furthermore, mice mono-colonized with the Δ*traA* strain cleared the infection ~25 days earlier than mice infected with wild-type AIEC (Fig. 3c). These data confirmed that type IV secretion is required for AIEC to persist in the host environment.

Biofilms are bacterial aggregates embedded in a matrix of exopolysaccharides, DNA, and protein that can protect bacteria from the immune system[55]. Given its known role in mediating bacterial aggregation[51], we considered the possibility that the T4SS facilitates AIEC persistence in the gut by promoting biofilm formation. We first quantified biofilm formation by AIEC on abiotic surfaces. Consistent with previous reports[51], the Δ*traA* mutant had a ~50% reduction in biofilm formation relative to the wild-type strain, indicating that TraA pili are required for bacterial aggregation (Fig. 4a). Some enteric pathogens can aggregate on the surface of host cells and form microcolonies[56]. Cellulose is a major component of the biofilm matrix in *E. coli* and was previously shown to mediate aggregation in AIEC[57,58]. We quantified AIEC microcolony formation on epithelial cells using immunofluorescence microscopy, defining a microcolony as the aggregation of more than 20 bacterial cells, which was the maximum number of bacteria we could detect in 99.8% of the aggregates formed by the cellulose-deficient mutant, Δ*bcsA* (Fig. 4b). Whereas wild-type AIEC readily deposited cellulose in microcolonies on epithelial cells, the Δ*bcsA* mutant did not (Fig. 4b). We compared the number of microcolonies formed by Δ*traA* and wild-type AIEC to examine the contribution of T4SS to this process. Although the Δ*traA* mutant adhered to and invaded epithelial cells to levels similar to wild type, the wild-type strain formed significantly more microcolonies on the host cell surface (Fig. 4c–f). The pronounced defect in microcolony formation by the Δ*traA* mutant was partially restored by

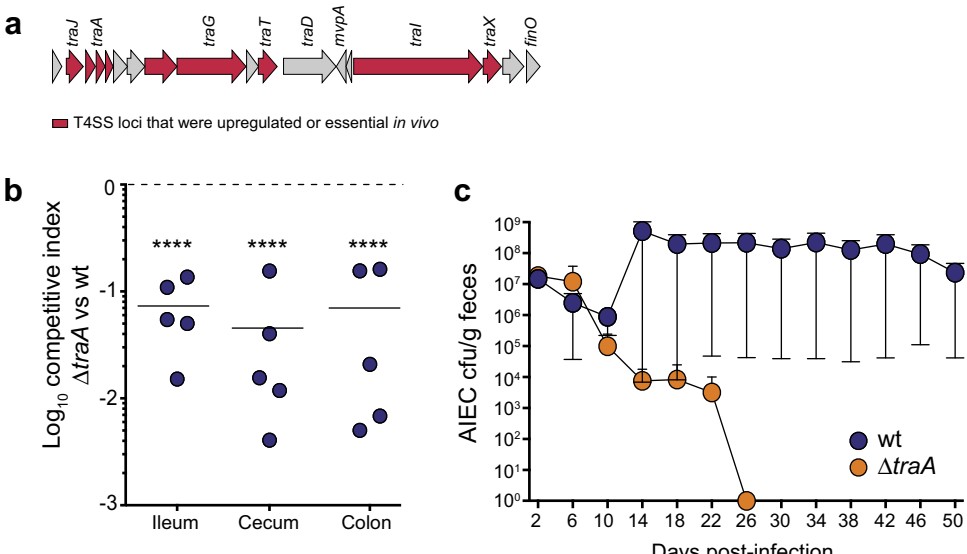

**Fig. 3 T4SS is required for AIEC persistence in the gut. a** Graphical representation of T4SS cluster in AIEC str. NRG857c. **b** Wild-type AIEC outcompeted the ΔtraA mutant in all gut tissues. Individual AIEC populations were enumerated in different tissues 12 days post-infection. Each data point represents the competitive index from an individual animal. Data presented are means ± SEM; n = 5 biologically independent replicates. Groups were compared against a value of "0" using Holm–Sidak method for multiple t-tests (two-tailed), ****P < 0.0001. **c** Fecal burden of different AIEC strains was monitored for 50 days. Results represented as colony forming units (cfu) per gram (g) feces. Data are means ± SEM; n = 5 biologically independent replicates. Groups were compared using two-tailed Mann–Whitney U test, P = 0.0005.

complementation of *traA* in trans. Because adhesion to host cell surfaces was not defective in the *traA* mutant, these results suggested that AIEC adhesion to epithelial cells may trigger T4SS-dependent aggregation. To investigate this hypothesis, we used RT-qPCR to measure the expression of multiple *tra* genes following AIEC contact with epithelial cells. Compared to planktonic AIEC, bacteria adhered to epithelial cells showed significantly greater expression of *traG, traD, traK, traBH*, and *traM* genes that are required for the assembly of T4SS (Fig. 4g). These results are consistent with the conclusion that the T4SS is involved in AIEC microcolony formation and biofilm formation on gut epithelial cells.

**AIEC activate T4SS and form microcolonies in the gut environment.** The data presented above suggested that host cells may promote surface aggregation of AIEC by providing an activating cue for T4SS expression. To track the expression of T4SS in vivo, we created a *tra-lux* transcriptional reporter[59] and measured in vivo luminescence following AIEC colonization.

Consistent with our RT-qPCR data from in vitro conditions that were free of host cells, we detected very low expression of *tra-lux* from AIEC growing on solid agar, whereas *lux* expression from a control constitutive promoter was readily observed on agar plates (Fig. 5a). When mice were infected with the *tra-lux* reporter strain, bioluminescence in the gut was pronounced by 24 h post-infection (Fig. 5b). These data are consistent with the notion that the intestine provides an inducing cue for AIEC T4SS expression. Since T4SS-deficient mutants had diminished colonization in our CD infection model, and the T4SS promoted AIEC microcolony formation on epithelial cells, we hypothesized that the T4SS promotes an alternative biofilm lifestyle in the gut that facilitates persistence. To test this, we infected mice with wild-type or ΔtraA AIEC strains and visualized the tissue-associated AIEC population by immunofluorescence microscopy 1 week after infection, a time point when the total bacterial burdens of the wild-type and ΔtraA mutant strains in the gut were similar (Fig. 3c). In mice infected with wild-type AIEC,

bacterial microcolonies were readily apparent on the epithelial surface in the cecum, whereas AIEC microcolony numbers were reduced by ~50% in mice infected with the ΔtraA mutant (Fig. 5c and d). Together, these data support the conclusion that AIEC use the T4SS machinery to form biofilms to persist in the gut environment.

**The T4SS is enriched in CD-associated E. coli.** To investigate the prevalence of T4SS in other CD-associated *E. coli*, we analyzed the genomes of 40 CD-associated *E. coli* isolates for the presence of *tra* genes using BLAST to determine whether they also contain T4SS[35,60–64]. As the query in our in silico analysis, we chose *traD*, the gene encoding for T4SS coupling protein, because *traD* is a homolog of *virD4* from the ancestral T4SS in *A. tumefaciens* and is a reliable marker to track the evolutionary history of T4SS in bacteria[17,65]. We queried *traD* from AIEC strain NRG857c against 40 published AIEC genomes using a threshold of 80% similarity and 90% coverage and aligned *traD*-containing gene clusters with the *tra* operon of AIEC NRG857c. Out of 40 CD-associated *E. coli* genomes interrogated, 12 had a T4SS-encoding cluster similar to the *tra* operon of AIEC strain NRG857c (Fig. 6a). Interestingly, one particular *E. coli* isolate, ST1, was isolated from the gut of a CD patient where it had persisted for 3 years and was isolated during peak inflammation[62]. Metagenomic analysis of the microbiome obtained from this patient revealed the presence of six other *E. coli* strains that appeared only transiently. Unlike the other *E. coli* strains, isolate ST1 displayed significant similarity to the AIEC pathovar and clustered with other CD-associated bacteria in the phylogenetic space[62]. Using BLASTn, we queried the seven total *E. coli* genomes obtained from this CD patient to search for *traD* homologs. Interestingly, only the chronically persisting strain, ST1, was positive for the *tra* operon, suggesting that the acquisition of T4SS provides a competitive advantage in the CD environment. To further investigate this possibility, we interrogated ~2700 human microbiome samples that were collected during various clinical studies for the presence of *tra* genes using

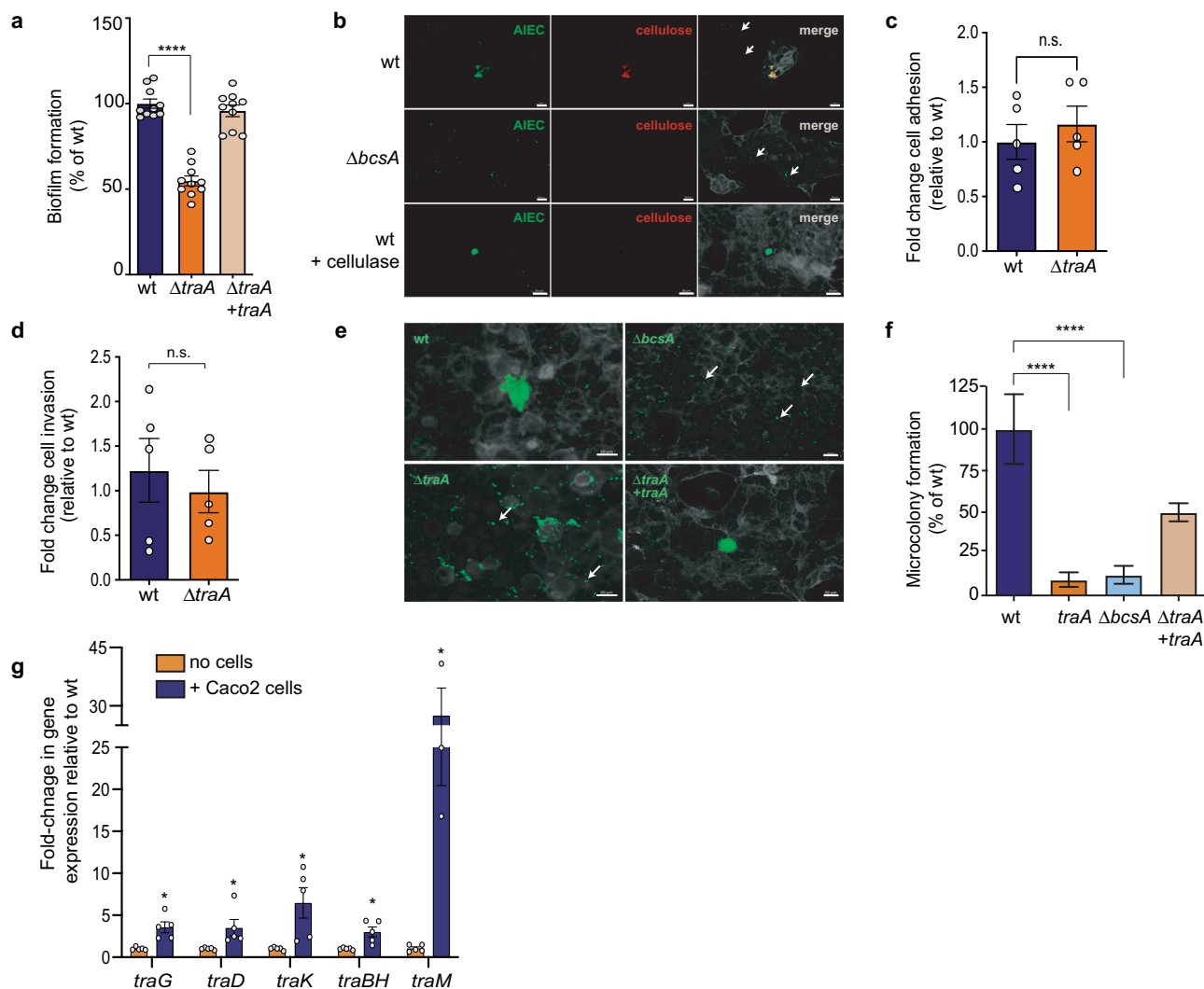

**Fig. 4 T4SS promotes biofilm formation by AIEC on epithelial cells. a** Biofilm formation by different AIEC strains on abiotic surfaces. Data are mean change ± SEM in the percentage of biofilm formation relative to wild type; $n = 10$ biologically independent replicates examined over three independent experiments. Groups were compared using two-tailed Mann–Whitney $U$ test, ****$P < 0.0001$ **b** Cellulose secretion by AIEC microcolonies. Immunofluorescence staining of Caco-2 cells infected with wild-type or Δ*bcsA* AIEC strains. Right panel, cellulose (red); middle panel, bacteria (green); left panel, merge of both panels showing Caco-2 cells (gray). White arrows indicate individual bacteria. Scale bar is 20 μm. Panels are representative of 15 images per group obtained from three biologically independent replicates. **c** *traA* is not required by AIEC to adhere to epithelial cells. Data are mean change ± SEM in adhesion relative to wild-type strain; $n = 5$ biologically independent replicates obtained from two independent experiments. Groups were compared using unpaired two-tailed *t*-test. **d** *traA* does not affect cell invasion by AIEC. Cell adhesion and invasion were quantified for the different AIEC strains indicated. Data are the mean fold change ± SEM in invasion relative to wild-type strain; $n = 5$ biologically independent replicates obtained from two independent experiments. Groups were compared using unpaired two-tailed *t*-test. **e** Immunofluorescence staining of different AIEC strains (green) showing microcolonies on epithelial cells (gray). White arrows indicate individual bacteria. Panels are representative of 20 images per group; $n = 3$ biologically independent replicates. Scale bar is 20 μm. **f** Microcolony formation was quantified using 20 fields of view per all groups except *bcsA* (23 fields of view); $n = 3$ biologically independent replicates. Data are the mean ± SEM microcolony formation relative to wild-type strain. Groups were compared using two-tailed Mann–Whitney $U$ test, ****$P < 0.0001$. **g** Fold change in gene expression of different *tra* genes in AIEC wild-type strain upon infecting Caco-2 cells; $n = 5$ biologically independent replicates ($n = 3$ for *traM*) obtained from two independent experiments. Groups were compared using Holm–Sidak method for multiple *t*-tests (two-tailed). Adjusted $P$ values are 0.016645 for *traG*, 0.03507 for *traD*, 0.033591 for *traK*, 0.027143 for *traBH*, and 0.010832 for *traM*.

MetaQuery, a public-use software for rapid quantitative analysis of specific genes within the human microbiome[66]. By normalizing the abundance of the queried genes to that of 30 universal single-copy genes, we could infer an average copy number for T4SS genes of interest across different gut microbiome samples. To increase the rigor of our analysis, we expanded the query space to include *traD* plus three additional T4SS markers, *traA, traG*, and *traI*, and calculated the relative abundance of *tra* homologs in the microbiomes of healthy donors and CD patients. This analysis revealed a significant enrichment of all *tra* homologs in the microbiomes of CD patients compared to the healthy controls (Fig. 6b). Together, these data suggest that the T4SS promotes the pathogenesis of CD-associated AIEC by facilitating their long-term persistence in biofilms in the host setting.

## Discussion

AIEC are recognized as a consistent microbial feature of CD; however, the genetic determinants of their pathogenicity and

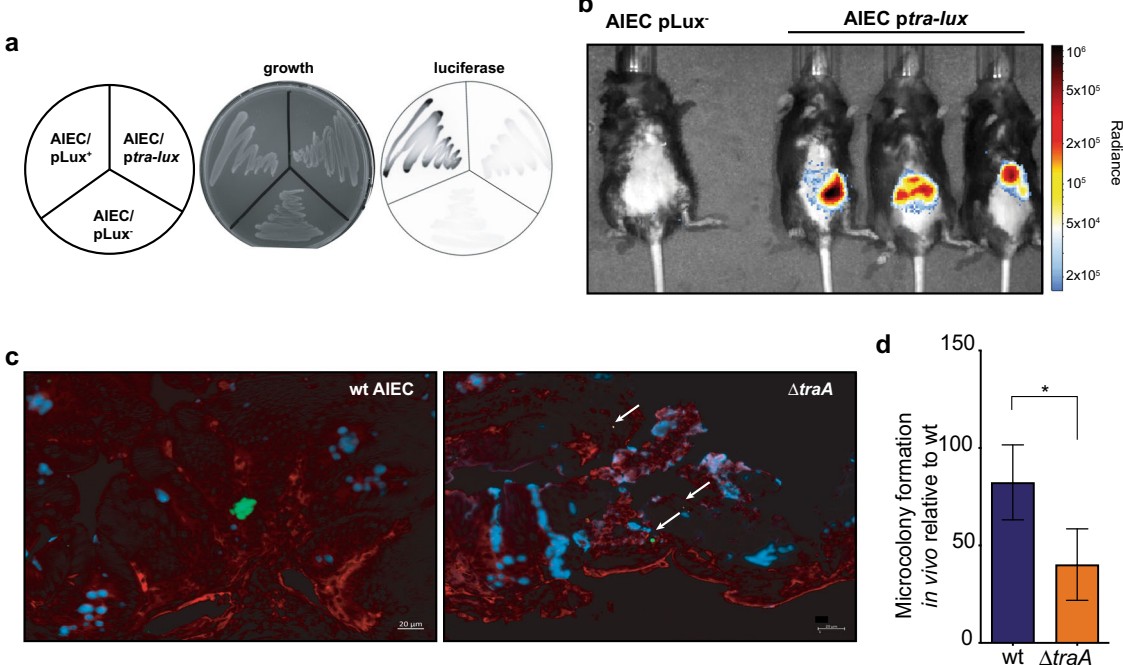

**Fig. 5 The host intestinal environment activates T4SS and mediates AIEC aggregation. a** Minimal expression of *tra-lux* transcriptional fusion in vitro. **b** T4SS is upregulated in the host environment. Representative whole-body images of mice infected with AIEC expressing *tra-lux* transcriptional fusion in the gut; *n* = 3 biologically independent replicates. Image is representative of two independent experiments. **c** Immunofluorescence staining of ceca from infected mice. AIEC (green, white arrows), mucin (blue), and actin cytoskeleton of intestinal cells (red); *n* = 3 biologically independent replicates and representative of two independent experiments. **d** AIEC microcolonies were quantified using 35 fields of view obtained from three biological replicates per strain. Presented is the mean ± SEM. Groups were compared using two-tailed Mann–Whitney *U* test, *P* = 0.0136; *n* = 3 biologically independent replicates and representative of two independent experiments.

chronic intestinal colonization are not fully delineated[10]. We conducted a high-density competitive fitness experiment using TIS to uncover the genetic determinants of AIEC colonization in a murine model of CD. Many of the genes that promoted AIEC colonization mapped to diverse pathways involved in the use of mixed gut metabolites. While the host and its microbiome often co-evolve toward mutual tolerance, CD-associated bacteria like AIEC can disrupt this ecosystem and exploit new metabolic resources that become available[10,67]. We found that AIEC is well-adapted to consume oxidized sugars produced during inflammation, including galactarate. Furthermore, our data indicate that the genes required for utilization of *Nos2*-derived metabolites are important for AIEC fitness in vivo. In addition to modulating the metabolic landscape in the gut, *Nos2* upregulation results in the accumulation of nitric oxide, which is toxic to bacteria[68]. The ability of certain bacteria to neutralize these toxic byproducts is crucial to their survival in vivo[30]. Indeed, our data revealed that genes involved in nitric oxide detoxification are required for AIEC fitness in the gut, strongly suggesting that metabolic diversity and immune evasion by AIEC are selective in the host setting.

The in vivo requirement for several two-component systems suggested that AIEC undergo a regulatory transition in the host environment. We compared the in vitro and in vivo AIEC transcriptomes to capture the changes in gene expression that were specific to the gut. The upregulation of the metabolic pathways required for the consumption of host sugars like *N*-acetyl glucosamine, propanediol, and galactitol were prominent features. Interestingly, mutations in the galactitol utilization operon were previously detected in several in vivo evolved AIEC lineages[9], suggesting a potential role for this metabolite in driving selection of pathogenic bacteria in the gut. In addition to

metabolism, our data highlighted the importance of iron acquisition to AIEC survival within the host. Several siderophore synthesis genes and transporters were upregulated in vivo, including the stealth siderophore, yersiniabactin, confirming and extending previous findings regarding the role of iron acquisition in AIEC virulence[48].

CD is marked by a decline in the microbial diversity in the gut and a simultaneous enrichment of *E. coli* strains[7]. This diminished taxonomic variation is accompanied by a less diverse metabolic landscape in the CD environment compared to the healthy gut[6]. Thus, microbes that can better compete for a more restricted metabolic profile would have a fitness advantage over members of the healthy microbiome. This is important, and might explain the sustained dysbiosis observed in the gut of CD patients, which is thought to contribute to disease progression[67,69,70]. In light of this, our data highlight the mechanisms employed by AIEC to outcompete commensals and disturb the gut ecosystem, which could provide a conceptual framework for gut remediation therapies that are currently being explored in CD.

By combining a multi-omic approach, we were able to resolve key elements of AIEC pathogenesis. Many in vivo essential genes identified by TIS were also significantly upregulated in the host environment, thereby strengthening the output. This included the T4SS, which is known to mediate biofilm formation in *E. coli* by promoting interbacterial adhesion[51]. We found that the T4SS was required by AIEC to form microcolonies on the surface of epithelial cells in a manner that was dependent on cellulose deposition. These results are congruent with previous work showing that cellulose secretion is vital to the formation of biofilms by *E. coli*[57,58]. Although type IV secretion is among the most ubiquitous secretion systems in bacteria, little is known

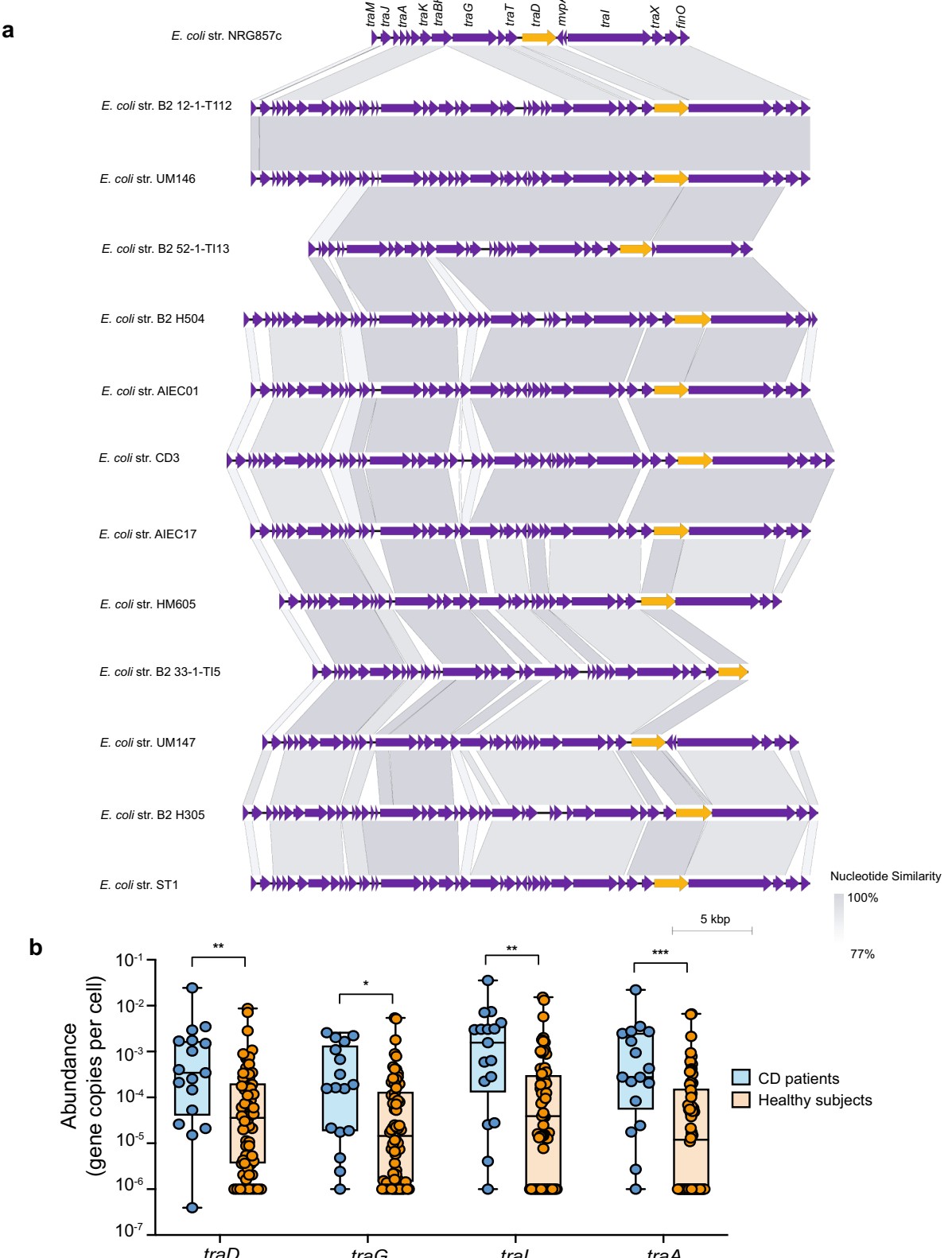

**Fig. 6 The prevalence of T4SS among CD-associated *E. coli*. a** T4SS clusters of different CD-associated *E. coli* strains aligned to AIEC str. NRG857c. The gray shading represents the nucleotide similarity (BLASTn) between the aligned regions. *traD* homologs are shown in orange. The scale bar indicates genomic length in Kbp. **b** Abundance of different *tra* genes in the microbiome of CD patients and healthy donors presented as gene copy number per cell; *n* = 17 CD patients and 66 healthy individuals whose data were obtained from MetaQuery webserver. Boxes extend from 25th to 75th percentiles and whiskers show the minimum and maximum values. Lines at the middle of each box show the median. Groups were compared using two-tailed Mann–Whitney *U* test and Bonferroni–Dunn method to correct for multiple comparisons. Adjusted *P* values are 0.008733 for *traD*, 0.028551 for *traG*, 0.001586 for *traI*, and 0.000277 for *traA*.

about the cues that activate the machinery[17]. We found that adhesion of AIEC to epithelial cells results in a significant upregulation of the T4SS cluster compared to planktonic AIEC. This expression pattern was not simply due to adherence to a substructure because the T4SS remained uninduced in AIEC growing on solid agar. In line with our suggestion that host-derived cues provide a signal for T4SS expression, we readily detected activation of the AIEC T4SS in the gut 1 day after infection of mice that allowed for persistent infection over chronic timescales. In accord with this, a mutant lacking the T4SS pili was outcompeted by the wild-type strain and showed significant deficiency in forming microcolonies and biofilms in vivo. Traditionally, biofilms have been implicated in protecting the bacterial population from exterior insults, like antibiotics and the immune system[58]. Thus, the formation of biofilms might allow AIEC to evade eradication by immune cells and thereby form a stable infection reservoir. Future work is required to decipher the specific contribution of microcolony and/or biofilm formation in promoting AIEC persistence in the host environment, and to uncover the specific host cue(s) involved in virulence factor induction by AIEC.

We queried the genomes of multiple AIEC strains and found that ~30% of them carry the genetic components to assemble a T4SS. Interestingly, an *E. coli* strain that persisted for 3 years in a CD patient, and isolated during peak inflammation, contained a T4SS that was similar to that found in NRG857c[62]. This is intriguing because it suggests a role for T4SS in mediating bacterial persistence in the CD environment. The frequent isolation of AIEC strains from CD patients suggests that the ability of these bacteria to persist in the gut environment is a common trait. A correlation between recurring flares of CD and *E. coli* abundance in the gut has been proposed[71]. Additionally, we detected an enrichment of *tra* homologs in the microbiomes of CD patients. While it is possible that these homologs are carried by AIEC strains, the power of our in silico approach to make this determination is limited by the lack of genetic markers for the AIEC pathogroup. We speculate that T4SS-mediated biofilm formation on the surface of epithelial cells establishes AIEC reservoirs for recurrent longitudinal and transmural dissemination. Further experimental work is required to investigate the contribution of AIEC biofilms to clinical flares that occur as part of the relapsing and remitting course of CD.

The complexity of the gut microbial community challenges our ability to determine the role of particular organisms driving CD[67]. Thus, delineating the mechanisms employed by specific gut bacteria as disease modifiers is of paramount importance to further our understanding of CD etiology. The approach to study AIEC virulence through a combination of high-throughput genome-wide screens to profile both the fitness requirements and transcriptome of a pathogen in the host environment proved highly effective. We identified a series of genes required for efficient gut colonization by AIEC that implicates exploitation of a defined metabolic network. Furthermore, this approach uncovered the role of T4SS in AIEC pathogenesis and showed that it is enriched among members of this pathovar compared to non-AIEC *E. coli*. Antibiotics are often used in CD management to eradicate the bacteria driving inflammation[72,73]. However, CD-associated *E. coli* are generally multi-drug resistant, limiting the therapeutic options available for treatment[73,74]. By delineating the genetic determinants important for host colonization, our results reveal new therapeutic targets for gut remediation strategies aimed at eradicating CD-associated pathobionts.

## Methods

**Mice**. Animal experiments were conducted according to Canadian Council on Animal Care guidelines using protocols approved by the Animal Review Ethics Board at McMaster University under Animal Use Protocol #17-03-10. Six to eight-week-old female C57BL/6 mice were purchased from Charles River. Animals were housed in a specific pathogen-free barrier unit under Level 2 conditions, temperature-controlled (21 °C), 30–50% humidity, 12 h light and dark cycle environment (dark from 7PM to 7AM) and were fed regular chow ad libitum.

**Growth of bacterial strains**. AIEC strain NRG857c and the isogenic strains were grown overnight at 37 °C in lysogeny broth (LB). When needed, antibiotics were added at the following concentrations: ampicillin 100 μg/ml, chloramphenicol 34 μg/ml, gentamicin 20 μg/ml.

**Bacterial strain construction**. To generate plasmid pSAM_SacB for transposon mutagenesis, the backbone of the parent plasmid pSAM_SDM[19] was amplified using TacTransposasefe NotI2 and pSAMfw4 primers (Supplementary Data 2). The *sacB* gene, encoding the toxic levansucrase, was amplified using SacBUpFw-NotI and SacBRv primers, pFLP2 as a template[75]. Both fragments were joined by overlap PCR and then cloned into the NotI site of pSAM_DGM. The AIEC strain NRG857c Δ*traA* mutant was generated via Lambda-Red recombination using the modified antibiotic selection system described before by our group[9,76]. Briefly, the knockout PCR product was amplified using primers traAKOFw and traAKORv (Supplementary Data 2) and pCDF_GmFrt as template. To select for the mutant, bacteria expressing the recombinase were transformed with the amplified product and plated on LB supplemented with ampicillin, chloramphenicol, and gentamicin at the concentrations mentioned above. The deletion was confirmed by Sanger sequencing. To generate an unmarked deletion, the gentamicin resistance cassette was excised using the FLP recombinase expressed from pKD46_km. Similarly, AIEC str. NRG857c *bla⁻* and *bcsA⁻* strains were generated using primers pO83blaKOFw/ pO83blaKORv and bcsAKOFw/bcsA-KORv, respectively. To generate the Δ*traA*/p*traA* complement strain, *traA* was amplified from the wild-type genome of strain NRG857c using primers SG03 and SG12 (Supplementary Data 2). Both the amplified product and pWSK129 were digested with EcoRI and SacI, followed by ligation to generate p*traA*. Similarly, p*tra*-Lux was generated by amplifying 500 bp upstream of the main *tra* cluster using primers wel317 and wel319 (Supplementary Data 2). Similarly, pGen_Lux was amplified using wel318 and wel320. The two fragments were joined using overlap PCR, followed by BamHI digestion and self-ligation to generate p*tra*-Lux.

**Mutant library construction**. To generate the AIEC mutant library, AIEC strain NRG857c Δ*bla* was grown overnight at 37 °C and mixed with the donor strain SM10/pSAM_SacB on LB agar supplemented with 1 mM isopropyl β-d-1-thiogalactopyranoside for 4 h to allow conjugation. To select for transposon mutants, a total of 100 conjugation mixtures were scraped, pooled, and resuspended in sterile 1× PBS, then plated on LB agar supplemented with chloramphenicol (34 μg/ml), gentamicin (20 μg/ml), and 2% sucrose to select against plasmid co-integrants. The next day, ~600,000 AIEC mutants were scraped off the LB agar and pooled together into LB broth supplemented with 20% v/v sterile glycerol, then stored at −80 °C for further genomic analysis.

**Mouse infections**. To screen the AIEC mutant library for in vivo fitness, three C57BL6/N mice were orally gavaged with 20 mg streptomycin the day before infection with 10⁹ CFU of the AIEC mutant library. Aliquots of the infection inoculum were taken for storage at −80 °C for later DNA extraction. Three days post infection, animals were euthanized, and their homogenized ceca were plated on LB agar supplemented with chloramphenicol (34 μg/ml) and gentamicin (20 μg/ml). Bacterial lawns were scraped and stored frozen at −80 °C for DNA extraction. For competitive infections, C57BL6/N mice were pre-treated with 20 mg streptomycin p.o. For infection, cultures of the strains of interest were grown overnight under selection. Mice were infected orally with a mixture of wild-type and Δ*traA* mutant strains. The inoculum was plated to obtain the input ratio between the two strains. Twelve days post-infection, animals were euthanized, and the infected tissues were homogenized in sterile 1× PBS and then plated on selective media to obtain the CFU per strain. Ratios were normalized to the input. To test the fitness of the Δ*traA* mutant in the gut, C57BL6/N mice were pre-treated with streptomycin as described above, then infected with 10⁹ CFU of AIEC str. NRG857c Δ*traA* mutant or wild type. Bacterial fecal burdens were measured by homogenizing feces in sterile 1× PBS followed by selective plating. For comparative transcriptomics, AIEC NRG857c was grown overnight in LB at 37 °C. The next day, the bacteria were sub-cultured and grown under the same conditions until stationary phase was reached. Bacterial pellets were harvested in Trizol and stored at −20 °C for later RNA extraction. Simultaneously, 10⁹ CFU aliquots of the culture were used to infect C57BL6/N mice for 3 days. At the end of the infection, mice were euthanized, and the ceca were retrieved. To harvest the bacteria, the ceca were opened longitudinally, and the mucus layer was gently scrapped into Trizol, then stored at −20 °C for later RNA extraction. To monitor the expression of the *tra* cluster in vivo, C57BL6/N mice were infected with 10⁹ CFU of the AIEC *tra*-lux strain. Mice were anesthetized (2% isoflurane carried in 2% oxygen) 24 h post-infection and imaged for 5 s in a IVIS Spectrum in vivo imaging system (Perkin Elmer).

**Sequencing and analysis of transposon mutant library**. DNA was extracted from bacterial pellets using a MasterPure DNA extraction kit (Epicentre), followed by digestion with MmeI restriction enzyme (NEB). The digested products were ligated to adapters adapt_t and adapt_b (Supplementary Data 2) followed by amplification using primers P7-NX and P5-X (Supplementary Data 2) with different indices to allow multiplexing. The amplified products were separated on 1% agarose gels, followed by gel extraction. The purified amplicons were sequenced on an Illumina HiSeq 2000 with single-end reads. The reads were examined for quality using FASTQC[77], then trimmed using Cutadapt[78] to retain the first 16 bp of the reads. Bowtie2 with default settings was used to map the reads to the reference genome of AIEC NRG857c. SAM files were used to tally the reads per "TA" site using previously published methods[18]. The complexity of each sequencing library was determined based on the number of unique "TA" sites disrupted in the genome. A multinomial distribution, derived from the mutant frequencies in the input library and scaled by the difference in complexity between the control library and the in vivo counterparts, was used to simulate control datasets that have the same number of total reads as the in vivo libraries to mimic the effects of genetic drift using previously published scripts[18]. Finally, the in vivo libraries were compared against the simulated control datasets using Mann–Whitney $U$ statistical test. One hundred Mann–Whitney $U$ tests were run to compare mutant abundance in each library to the corresponding 100 simulated libraries. The average $P$ value from the 100 Mann–Whitney $U$ tests is reported for each locus in each of the three biologically independent replicates (Supplementary Data 1). Mutants showing <0.5-fold reduction in abundance in all three biological replicates compared to the in vitro control ($P < 0.01$) were considered to be significantly attenuated in colonizing the host. All scripts used are available as part of the ARTIST pipeline[18]. All genomic loci, including those that showed significant variation in abundance in vivo, were functionally annotated using the webtool, eggNOG-mapper 4.5.1 (ref. [79]).

**RNA-sequencing and differential expression analysis**. RNA was extracted using a Trizol-based method. Briefly, gut samples were homogenized in 900 μl Trizol (Ambien) using a metal bead. Chloroform was added to extract proteins, then samples were centrifuged, and the nucleic acid was precipitated from the supernatants using isopropanol, then resuspended in nuclease-free water. The quality of the purified RNA was verified with an Agilent RNA 6000 Nano Bioanalyzer. Only samples with RNA integrity numbers ≥7 were used for further processing. Ribosomal RNA was depleted using Ribo-Zero (Illumina), followed by the generation of barcoded cDNA for each sample. cDNA was sequenced on an Illumina HiSeq 2000 platform with single-end reads. The quality of the raw reads was confirmed using FASTQC[77]. Reads were trimmed using Cutadapt[78] to remove Truseq adapter sequences, then mapped to the host reference genome GRCm38 using BWA (mem algorithm)[80] with default settings to remove non-bacterial sequences. Next, reads were mapped to the reference genome of AIEC NRG957c using BWA with default settings. Mapped reads were tallied using FeatureCounts[81] and the differential gene expression was determined using the R package DESeq2 (ref. [34]). Genes were considered differentially regulated if they showed log₂ fold change >0.5 or < −0.5 at FDR-adjusted $P$ value <0.05. To generate the heatmap, transcripts per million (TPM) values of the significantly regulated genes were calculated for each replicate. Values were scaled by row then used to generate the heatmap using Pheatmap 1.0.10 R package[82].

**Abiotic biofilm assay**. AIEC strains were grown overnight at 37 °C. Overnight cultures were diluted 1:200 in LB no salt, then the cultures were incubated statically in flat bottom polystyrene plates (VWR) for 24 h at 37 °C to allow for biofilm formation. At the end of the incubation period, bacterial growth was measured by reading absorbance at 600 nm. Wells were then washed twice with deionized water to remove media and non-adherent bacteria, then stained with 0.1% (w/v) crystal violet (Sigma-Aldrich) for 10 min. To remove excess crystal violet, two additional washes with deionized water were completed. Crystal violet-stained biofilms were solubilized in 95% (v/v) ethanol. Absorbance was measured at 595 nm using an EnVision 2104 Multilabel Reader. OD595 readings were divided by the optical density of the cultures to obtain normalized biofilm production[83].

**Immunostaining of infected cells and tissues**. Caco-2 cells were grown in Dulbecco's Modified Eagle's Medium (DMEM, Gibco) supplemented with 20% fetal bovine serum (Gibco), 1× minimum non-essential amino acids (Gibco), 10 units/ml penicillin, 10 μg/ml streptomycin (Gibco), and 1 mM HEPES (Gibco). Caco-2 cells were seeded on poly-L-lysine-coated cover slips and were grown to ~90% confluency. The cells were infected with different AIEC strains at a multiplicity of infection (MOI) of 25:1 for 1.5 h. Non-adherent bacteria were washed off three times with 1× PBS, and the media was changed to Iscove's Modified Dulbecco's Medium (Gibco). After 6 h, the media was removed, and the cells were washed three times with 1× PBS. Samples were fixed with 4% paraformaldehyde (Sigma-Aldrich) for 20 min at room temperature. Coverslips were blocked overnight at 4 °C in 3% goat serum (Invitrogen) and 1% bovine serum albumin (Sigma-Aldrich). To stain AIEC, samples were incubated with primary antibody anti-O83 (1:300 dilution, Statens Serum Institute, CAT: 85077). When needed, coverslips were incubated with 0.005% calcofluor white stain (Sigma-Aldrich) for 20 min at room temperature to visualize cellulose. Coverslips were washed three times with

1× PBST, followed by staining with Alexa Fluor 488 goat anti rabbit (1:200 dilution, Invitrogen, CAT: A-11034) and phalloidin 568 (1:40 dilution, Invitrogen, CAT: A12380) for 40 min at room temperature. To degrade cellulose produced by AIEC in the exopolysaccharide matrix, fixed cells were treated with 100 units of cellulase (Sigma-Aldrich) for 2 h at 37 °C, then stained as mentioned above. To stain AIEC in infected tissues, C57BL/6 mice were orally gavaged with 20 mg of streptomycin. After 24 h, animals were infected with 10⁹ CFU of AIEC wild-type and ΔtraA strains. For each strain, three animals were used. Five days post-infection, mice were euthanized, and the ceca were fixed in carnoy solution (60% ethanol, 30% chloroform, and 10% glacial acetic acid) overnight at 4 °C, followed by washing and storage in 70% ethanol before paraffin embedding. Five-micrometer sections were cut onto positively charged glass slides. Next, samples were de-paraffinized with xylene, and treated with pre-warmed antigen retrieval solution (10 mM citric acid, pH 6) for 40 min. To stain for mucin, slides were incubated with wheat-germ agglutinin in (1:200 dilution, 1% goat serum, 0.1% BSA, Invitrogen, CAT: W11263) for 1 h at 37 °C. Next, samples were permeabilized with 0.1% Triton X-100 in PBS, then incubated with rabbit anti-O83 (1:300 dilution), and mouse anti-β actin (1:500 dilution in 1% goat serum, 1% Tween 20, 0.1% BSA, Abcam, CAT: ab8226) for 40 min at room temperature. Next, samples were washed with PBS, then incubated with anti-rabbit Alexa-Fluor 488 (1:200 dilution, Invitrogen, CAT: A-11034), and anti-mouse Alexa-Fluor 568 (1:200 dilution, Invitrogen, CAT: A-11031) as the secondary antibodies for 1 h at room temperature. All slides were examined on a Zeiss Axio Imager with 20× and 40× objectives. Images were captured with a Hamamatsu Digital camera ORCA-R2. Samples were examined blindly. Images were processed with ImageJ software (version 1.51) for bacterial quantification. In our microscopy setting, the surface area of one E. coli bacterium is 138 sq. pixels. Images were obtained from three independent experiments.

**RT-qPCR**. To track the expression of *tra* genes in the adherent AIEC population, wild-type AIEC was grown in LB at 37 °C overnight. The next day, bacteria were used to either infect Caco-2 cells at an MOI 1000:1 or cell-free DMEM media for 4 h. Importantly, non-adherent bacteria were washed out with sterile PBS. At the end of the infection, bacterial pellets were harvested from both conditions and nucleic acid was extracted using the Trizol-based method. Upon treatment with RNase-free DNase (Thermo Fisher), cDNA was obtained using either qScript cDNA Supermix (Quantabio) or protoscript (NEB). qPCR was performed with SYBR green SuperMix (Quanta Biosciences) on the LightCycler 480 (Roche) or Bio-Rad CFX96 real-time PCR detection systems. Primers used are listed in Supplementary Data 2. Data analysis was performed using the LightCycler software (version 1.5). Gene expression was normalized to the housekeeping gene, *gyrB*.

**Cell adhesion and invasion assays**. Adhesion assays for AIEC were performed using standard methods[84]. Briefly, Caco-2 cells were infected with different AIEC strains at MOI 10:1. After 1.5 h, cells were washed with sterile 1× PBS three times to remove non-adherent bacteria, then lysed using 1% Triton X-100 in PBS. To quantify cell invasion, cells were infected for 1.5 h. Following the infection, the cells were washed three times with sterile PBS then incubated with DMEM supplemented with gentamicin (100 μg/ml) to kill extracellular bacteria. After 1 h of incubation with gentamicin-supplemented media, cells were washed three times with sterile PBS, then lysed with 1% Triton X-100 in PBS. All lysates were serially diluted and plated on selective media. Results were obtained from three biological replicates.

**Comparative genomic analysis of T4SS**. Genome assemblies for different CD-associated E. coli strains were obtained from NCBI and the repository of the Australian National University. *traD* of AIEC str. NRG857c was queried against the genomes of the different strains using BLASTn search with a threshold of 80% similarity and 90% coverage. T4SS clusters were extracted from contigs carrying *traD* homologs. Sequence alignment was done using BLASTn[85,86] and visualized using EasyFig version 2.2.2 (ref. [87]). To interrogate the human microbiome database for the presence of *tra* homologs, we queried four *tra* genes, *traADGI*, using the MetaQuery web server using 70% coverage for both the query and the target, and a minimum identity of 70%, with maximum e-value of 10⁻⁵ (ref. [66]). For samples obtained from the same subject, gene copy numbers were averaged to avoid redundancy.

**Statistics and reproducibility**. Unpaired $t$-tests or two-tailed Mann–Whitney $U$ tests with Bonferroni–Dunn or Holm–Sidak methods to correct for multiple comparisons were performed at 95% confidence interval to determine the difference among groups. Analyses were performed using Graph Prism 9.0 (GraphPad Software), DESeq2 (ref. [34]), or MATLAB version R2017b. A $P$ value of <0.05 was considered significant for all analyses, except TIS analysis in which a $P$ value of <0.01 was the cut-off for significance. The number of subjects tested in each experiment is indicated in the figure legends or within the individual figure panel. Microscopy images are representative of at least two independent experiments.

**Reporting summary**. Further information on research design is available in the Nature Research Reporting Summary linked to this article.

## Data availability

Sequence data that support the findings of this study have been deposited in Sequence Read Archive (SRA) database and are accessible through the SRA accession number PRJNA684126. RNA-sequencing data that support the findings of this study have been deposited in National Center for Biotechnology Information Gene Expression Omnibus (GEO) and are accessible through the GEO Series accession number GSM4969708. All other relevant data are available from the corresponding author upon reasonable request. Source data are provided with this paper.

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

## Acknowledgements

W.E. was supported by a postdoctoral fellowship from the Canadian Institutes of Health Research. A.M.O. and C.N.T. were supported by Canada Graduate Scholarships from the Canadian Institutes of Health Research. B.K.C. holds the Canada Research Chair in Infectious Disease Pathogenesis. This research was funded by grants to B.K.C. from the Canadian Institutes of Health Research (324932) and Crohn's and Colitis Canada. We thank Lauren and Benjamin Coombes for their kind assistance with this project.

## Author contributions

Conceptualization, W.E. and B.K.C.; methodology and investigation, W.E., S.H., S.G., A.M.O., C.N.T. and T.P.H.; data analysis and curation, W.E.; writing—original draft, W.E. and B.K.C.; writing—review and editing, W.E., B.K.C., and M.K.W.; supervision, B.K.C.; funding acquisition, B.K.C.

## Competing interests

The authors declare no competing interests.
