## [Peer Review File · Nature Communications]

REVIEWER COMMENTS

Reviewer #1 (Remarks to the Author):

This is very well-written manuscript focused on identifying fitness genes in Crohn's disease-associated pathobionts adherent-invasive *E. coli* (AIEC). I enjoyed reading the manuscript.

In this manuscript, Elhenawy et al used transposon mutagenesis and bacterial transcriptome approaches and identifies genes responsible for in vivo fitness of a strain of AIEC. The authors first employed transposon mutagenesis and identified various metabolic pathway genes (e.g., nitrate respiration, amino acid metabolism, fucose utilization, mucin glycan foraging). Next, using in vivo RNA-seq approach, the transcriptional changes in AIEC in the gut were examined. By analyzing the results obtained from these two approaches, the authors determined the genes that are up-regulated during adaptation to the gut and required for the in vivo fitness. Among identified 149 genes, the authors focused on genes encoding type IV secretion system (T4SS), the *tra* operon. The authors demonstrated that T4SS is required for the colonization and persistence of AIEC in the gut. As a mechanism, the authors found that T4SS is crucial for the formation of biofilm. Lastly, the authors demonstrated that T4SS is conserved in strains of AIEC and enriched in the microbiome of CD patients compared to healthy control subjects.

Although AIEC are implicated in CD pathogenesis, there are no genetic determinants that discriminate AIEC from commensal *E. coli*. Hence, identifying unique pathways used by AIEC but not by commensals would lead to new therapeutic strategies that selectively eradicate AIEC. Therefore, this study is important and has the potential to result in new therapies for CD. Overall, this is a very well-performed study, but this reviewer feels some points need to be revised to improve the quality of this work.

Specific comments:

1. The authors nicely performed the Tn mutagenesis approach and identified various metabolic pathway genes that are responsible for the fitness of AIEC. However, some of those pathways (e.g., nitrate respiration) are commonly used by bacteria belonging to the Enterobacteriaceae family, including non-pathogenic, commensal *E. coli*. It would be more informative if genes/pathways that are selectively used by AIEC, but not commensal *E. coli*, are identified.
2. For the TIS screening, streptomycin-treated mice were used. In this condition, genes/pathways that are required for the competition between AIEC and commensal bacteria cannot be identified. It would be interesting if the authors could perform TIS screening without depleting commensal competitors.
3. Does T4SS contribute to the inflammatory properties of AIEC besides biofilm formation? What about the adherent/invasive capacity of *traA* mutant? Any defect in the induction of pro-inflammatory cytokines from epithelial and/or immune cells?
4. In Fig. 3c, the colonization of WT AIEC was even enhanced from day 14. In contrast, the colonization of the *traA* mutant started to decline from this time point. Since the differences in the gut colonization between WT and the mutant were minimal until this time point, T4SS is required to resist events that occurred around 2 weeks post-colonization. What has happened at this time point? Inflammation becomes evident around 14 days post AIEC infection? Adaptive immune responses against AIEC (e.g., specific T effector cells, IgA) are developed? Assessing intestinal inflammation (e.g., histology, fecal lipocalin-2 level) or AIEC-specific immune responses (e.g., AIEC-specific IgA, Th17) would help support the claim that T4SS renders the pathogen resistant to inflammation and/or host immunity. Can the *traA* mutant persist in the gut of Rag-deficient mice (lacking AIEC-specific T/B cell immunity)?

Reviewer #2 (Remarks to the Author):

The study by Elhenawy et al. uses an in vivo transposon insertion sequencing screen to identify a cluster of T4SS related genes that are required for AIEC pathogenesis. Overall the experiments are well designed, and the conclusions are justified by the data provided.

Major points

- 1) Since there is a traA complement experiment in Figure 4, it would be nice to see if the competitive index and fitness advantage in Figure 3b and 3c are reduced when using a traA/ptraA complement strain.
- 2) Figure 4f lacks names of the different tra genes referenced in lines 286-287.
- 3) Lines 340-342: Does the higher abundance of tra homologs in patient indicate enrichment of tra encoded AIEC's in CD patient samples or just enrichment of AIEC's in CD patient samples? The way the data is presented does not separate between these two interpretations. It is unclear how "cell" in gene copies per cell is defined.

Reviewer #3 (Remarks to the Author):

Elhenawy et al, aim to determine the mechanism by which adherent-invasive E. coli (AIEC) act as an active disease modifier in Crohn's disease (CD). The authors use Transposon-insertion sequencing (TIS) to conduct an analysis of gene essentiality and/or dispensability in the colonization of the murine gut (cecum). They provide examples of functions they know to be essential/important in vivo. They use RNA-Seq to explore AIEC regulatory changes in vivo by comparing it to an in vitro expression profile. Among the upregulated genes they provide examples of those involved in various forms of metabolism. By comparing TIS and RNA-Seq data they aim to further/again determine genes that mediate functions critical for AIEC pathogenesis and/or colonization. In the 2nd part of the paper the authors focus on a type IV secretion (T4SS) system that was found to be important for survival in vivo. The role of T4SS in biofilm and microcolony formation is studied and the prevalence of T4SS/tra-genes is associated with CD. Overall the study is interesting and has some neat data, however the first half of the study is problematic due to the lack of statistics supported analyses (of either TIS or RNA-Seq data) that have a clear idea/goal in mind. Additionally, the authors only provide some of the data and in a format that makes it inaccessible for anyone that wants to redo or build on the analyses that were done, or simply explore the data that were generated. Moreover, there is little coherence between the first and second part of the study; i.e. there is no clear rationale/analyses that leads from the first to the 2nd part (see for instance point 17 below). All in all, I think the work has some real potential, but right now the two parts feel like they are each a part of a different paper, giving the overall idea that the work is half finished.

Major concerns:

1. The authors describe that "...347 genetic loci [were identified] that enabled AIEC colonization of the host yet were completely dispensable in vitro (Figure 1C)." It is unclear what comparison is made here, e.g. how was it determined that these 347 loci were completely dispensable in vitro? If compared to in vitro data, then what was the comparison exactly? And if it was compared to in vitro was the in vitro sample also re-plated and exposed to gentamycin and chloramphenicol like the in vivo selected libraries? If not than how is this difference in treatment evaluated to affect the in vivo selected library? It is also impossible to assess the amount of variance in the data and the bottlenecks that likely occurred in the experiments. In that sense 3 mice also sounds like a very small number of mice. How stable/confident are the results really (also see point 6)?
2. All TIS data should be provided. Including raw read numbers, ratios of input/output, numbers of insertions for each insertion in each genetic region. Without these data it is impossible to recreate and/or assess any of the analyses. These data should be configured in easily accessible files and

formats. For instance, a single excel file or csv file. These files should at the least contain each insertion for all genes in the genome, their annotation and gene categories (also see below). Without it, the data are not accessible/usable for anyone else. One of the points of publishing a study like this is that other people can assess and use the data for their own experiments/ideas. This is impossible in the current format.

3. Transposon construction: since the authors are presenting a new 'tool', it would be useful to provide a detailed annotated plasmid map, instead of only a schematic.

4. Related to comments 1 and 2, since only a very small part of the data/analysis is provided it's impossible for anyone to re-run, re-analyze or appropriately interpret the data for instance in Table S1, which makes it hard to determine what the results mean.

5. Fig 1D lists functional assignments. I can't find those functional assignments associated with genes COGs, gene names and data anywhere. Thereby one can't assess and/or rerun the analysis, adjust it or use the data for one's own study.

6. P5-6L110-122: The authors seem to do a lot of cherry picking, i.e. highlighting a bunch of genes that have low abundance, which confirms genes that are known to be important under these or similar conditions. While such highlighting is fine and important. It is only a first step and it certainly does not validate the genomic approach. Proper validation would include a significant number of genotype-phenotype experimental validations in the mouse gut, only than we can assess the real strength and confidence of the data/approach/analysis.

7. A STRING network is presented in FigS2. It is unclear how this network was extracted from the database as the required information to do this is not presented, i.e. I couldn't find it. It is thereby unclear how the STRING network was generated. Importantly what is the value of the STRING network? There is no real network-analysis here (P27L512-514: This does not describe/define a network analysis). The network could be valuable just for data organization, and while it looks cool its value here is not clear.

8. P7L125-127. "The COG analysis...". What COG analysis is performed here? If a proper COG analysis is performed, what then are the statistics or enrichment scores? There could definitely be enrichment here, and if you write analysis then do the statistics. If not then say there's a trend or a tendency for something. I'm fine with the authors just cherry picking some data here that helps support some ideas, but than just say that that's what you're doing. But preferably do some proper analysis, like an enrichment analysis, which can have its own shortcomings, but there are several different options to do some good analyses for pattern discovery.

9. P7-8L125-150. Again, a lot of what the authors write about seems to be cherrypicked from the data, while there don't seem to be any ideas or hypotheses that are generated from any specific analysis. Also there doesn't seem to be any unifying and/or new theme that emerges from these descriptions. The presented 'analysis' thereby doesn't add any value on top of the generated data. It is evident that things are/will be different in vivo, however the 'analysis' does not create much order in the data that gets presented.

10. P8L167 "Our TIS screen revealed the presence of several two-component systems that are required for AIEC fitness in vivo, suggesting that AIEC undergo regulatory adaptation to the host environment." Indeed, I don't think anyone will deny things change in vivo. But this feels like an example of Chekov's gun : ' If in the first act you have hung a pistol on the wall, then in the following one it should be fired'. However, nowhere in the proposal do these two component systems (TCS) return to fulfill a role. So why so much emphasis on these TCSs?

11. P8L170: "Three days after AIEC infection, total RNA was isolated from the ceca of mice and sequenced to characterize AIEC gene expression (Figure 2A)."
What was the quality of the RNA isolated from these in vivo samples? RNA-quality from in vivo samples can be wildly different and of bad quality, affecting any downstream analysis. How does the quality relate to the in vitro quality? How was this assessed? How did Streptomycin treatment

affect transcription? What was the in vitro condition to generate in vivo DE genes? Was there an in vitro Spectinomycin control to assess the effects of Spectinomycin? What was the growth phase of the in vitro sample and how does growth phase affect the observed in vivo DE genes?

12. Similar to the TIS data above, where are the raw read numbers for the RNA-Seq data, the processed numbers (e.g. featurecounts tables, DEseq files, annotation and COGs?), so people can actually easily use and evaluate it? The data is just not easily accessible and/or unexplorable right now. Overall no effort seems to have gone into making any of these data (TIS/RNA-Seq) easily explorable and accessible, which goes against the FAIR principles and which is a shame.

13. I couldn't find the data the venn diagram is based on. Which genes are in the overlaps and non-overlaps? What are their functions, COGs, values, how do these data/types relate to each other?

14. Overall Figures 1 and 2 are largely uninformative. There is not much one would take away from it.

15. P10L187 "Among these, 1,634 genes could be assigned to 17 COGs, with nearly half (46%) belonging to metabolic pathways, consistent with the TIS data that revealed the importance of metabolic adaptability." Again, this would require a proper statistical/enrichment analysis to show that this suggestion is actually supported. For instance, if a large proportion of your genome is associated with metabolism, then a large proportion of DE genes could naturally be associated with metabolism. And again, no reader would be able to do this analysis either, because the proper files/info are missing.

16. P10L192 – 218 feels like cherry picking. No real computational/network analysis here, with no validation, and no way to easily find these results in the data. The authors suggest interesting things, but up to now it's all observations, that do not contribute to a coherent story.

17. P11L223. "To uncover new genes and pathways that contribute to AIEC infection biology, we cross-referenced the list of genes that were differentially regulated in the gut to the set of genes that were required for in vivo growth identified by TIS. Our rationale was that genes essential for gut colonization and also differentially upregulated upon entering the gut environment likely mediate critical functions in AIEC pathogenesis and colonization."

This is not a rationale that holds water. It is unclear why a TIS gene (i.e. one with a fitness defect) that is also differentially regulated would be more important than a non-differentially regulated TIS gene. There are plenty of studies that show a clear disconnect between genes important for fitness, and those that are transcriptionally regulated. There are also several examples of analyses that can combine these data types and then help extract important patterns from the data. The problem with 'the rationale' is that it suggests that the authors used some statistical/network analyses to get to the next and most important part of the paper, which focuses on the T4SS. In reality the presented analyses in the 1st part does not lead to the 2nd part of the paper. The most important problem here is that there is no clear reasoning why the T4SS is picked out of the large number of TIS genes and suddenly becomes the focus of the data-set and paper. There is nothing wrong with focusing on the T4SS, I think it's a very interesting phenotype that gets presented in the remaining part of the paper, however the presented TIS and RNA-Seq data in the first half of the paper overall have very little, to nothing to do with the second part of the paper. The authors could basically have started the paper with "T4SS was identified in a TIS screen as important for survival in the cecum." And then only present the 2nd half of the paper. Right now, in the way the 1st half of the paper is analyzed and presented, I do not see the added value to the manuscript.

18. P12-Table S3 –This table does not contain clearly presented cross-referenced- TIS/RNA-Seq data.

19. P12L228. "This analysis identified 149 genes that met these criteria, with the majority of them controlling metabolic functions, confirming the pivotal role of metabolic adaptation in promoting AIEC in vivo fitness." – Again I don't think it does, you would need to do the proper analysis to show its different from the distribution you'd expect based on the genome distribution.

20. P11L229 – 235 – Subsequently, as above in points 9 and 16 these are more ad hoc explanations for some of the data, which does not equal proper analysis

21. P13L265 and P17L314. It is unclear whether microcolonies are the same or similar to biofilms. The importance of microcolonies in the infection is also not clear; they may be better at resisting antibiotic mediated clearance, or maybe they provide an advantage during invasion of resident populations. This is a missing link in the paper; i.e. what is the true advantage of the phenotype? On P19L342. The authors state: "Together, these data suggest that the T4SS promotes the pathogenesis of CD-associated AIEC by facilitating their long-term persistence in biofilms in the host setting." Also here biofilms may not be the right wording as there is mostly evidence for the importance of microcolony formation.

22. P19Fig6B. This analysis is not clear and no (raw) data is provided to redo/rework/assess the analysis; e.g. which genomes were selected, what is the associated metadata? While the finding/association between tra-genes and CD is interesting, it is not possible to assess whether the comparison/association that was made is fair.

REVIEWER COMMENTS

Reviewer #1 (Remarks to the Author):

This is very well-written manuscript focused on identifying fitness genes in Crohn's disease-associated pathobionts adherent-invasive E. coli (AIEC). I enjoyed reading the manuscript.

In this manuscript, Elhenawy et al used transposon mutagenesis and bacterial transcriptome approaches and identifies genes responsible for in vivo fitness of a strain of AIEC. The authors first employed transposon mutagenesis and identified various metabolic pathway genes (e.g., nitrate respiration, amino acid metabolism, fucose utilization, mucin glycan foraging). Next, using in vivo RNA-seq approach, the transcriptional changes in AIEC in the gut were examined. By analyzing the results obtained from these two approaches, the authors determined the genes that are up-regulated during adaptation to the gut and required for the in vivo fitness. Among identified 149 genes, the authors focused on genes encoding type IV secretion system (T4SS), the tra operon. The authors demonstrated that T4SS is required for the colonization and persistence of AIEC in the gut. As a mechanism, the authors found that T4SS is crucial for the formation of biofilm. Lastly, the authors demonstrated that T4SS is conserved in strains of AIEC and enriched in the microbiome of CD patients compared to healthy control subjects.

Although AIEC are implicated in CD pathogenesis, there are no genetic determinants that discriminate AIEC from commensal E. coli. Hence, identifying unique pathways used by AIEC but not by commensals would lead to new therapeutic strategies that selectively eradicate AIEC. Therefore, this study is important and has the potential to result in new therapies for CD. Overall, this is a very well-performed study, but this reviewer feels some points need to be revised to improve the quality of this work.

Specific comments:

1. *The authors nicely performed the Tn mutagenesis approach and identified various metabolic pathway genes that are responsible for the fitness of AIEC. However, some of those pathways (e.g., nitrate respiration) are commonly used by bacteria belonging to the Enterobacteriaceae family, including non-pathogenic, commensal E. coli. It would be more informative if genes/pathways that are selectively used by AIEC, but not commensal E. coli, are identified.*

Response: We thank the reviewer for their careful read and positive feedback. We have previously shown that the metabolic behavior of AIEC in the gut is different from that of commensal *E. coli* (Elhenawy *et al.* Cell Host & Microbe, 2019). Our current study lends further support to this notion. Also, previous work by Dong-Eun Chang *et al* (2004) showed that amino acid catabolism is not important for commensal *E. coli* to colonize the mouse intestine, whereas our data underscore an important role for amino acid metabolism in the ability of AIEC to colonize the gut environment. We have included this new reference in the revised manuscript to help place our work in context within the field. Thus, our study opens new avenues to disentangle the differences between AIEC and commensals. Future work could involve repeating the TIS experiment with commensal strains of *E. coli* to further identify the metabolic requirements that distinguish the two *E. coli* groups, but this was not our focus for this work and our experiments were not designed for this type of comparison.

2. *For the TIS screening, streptomycin-treated mice were used. In this condition, genes/pathways that are required for the competition between AIEC and commensal bacteria cannot be identified. It would be interesting if the authors could perform TIS screening without depleting commensal competitors.*

Response: We thank the reviewer for this interesting comment; however, it is not technically possible to carry out the proposed experiment. Without this antibiotic treatment the infection bottleneck is too tight,

maximizing the risk of amplifying the negative effects of genetic drift. It might be possible for one to address this question using germ-free mice, but this was outside the scope of our study.

3. Does T4SS contribute to the inflammatory properties of AIEC besides biofilm formation? What about the adherent/invasive capacity of *traA* mutant? Any defect in the induction of pro-inflammatory cytokines from epithelial and/or immune cells?

Response: We thank the reviewer for this suggestion. We showed that adherence to epithelial cells is not affected by *traA* deficiency (original Figure 4c). As requested, we completed the suggested experiment to investigate invasion, which reinforces the adhesion data and demonstrates that invasion is also not affected by the loss of *traA* (new Figure 4d).

4. In Fig. 3c, the colonization of WT AIEC was even enhanced from day 14. In contrast, the colonization of the *traA* mutant started to decline from this time point. Since the differences in the gut colonization between WT and the mutant were minimal until this time point, T4SS is required to resist events that occurred around 2 weeks post-colonization. What has happened at this time point? Inflammation becomes evident around 14 days post AIEC infection? Adaptive immune responses against AIEC (e.g., specific T effector cells, IgA) are developed? Assessing intestinal inflammation (e.g., histology, fecal lipocalin-2 level) or AIEC-specific immune responses (e.g., AIEC-specific IgA, Th17) would help support the claim that T4SS renders the pathogen resistant to inflammation and/or host immunity. Can the *traA* mutant persist in the gut of Rag-deficient mice (lacking AIEC-specific T/B cell immunity)?

Response: We thank the reviewer for these comments and interesting suggestions. While these questions are of interest to us, the focus of this work was not on the mucosal immune response to AIEC. The experiments with Rag-deficient mice would be interesting, however we believe that the negative impact of Rag deficiency on immune development in the gut, and the complicating factor of different microbiota in such mice, would confound the interpretation of these experiments. To address this question properly would require an experimental approach well beyond the scope of this current study.

Reviewer #2 (Remarks to the Author):

The study by Elhenawy et al. uses an in vivo transposon insertion sequencing screen to identify a cluster of T4SS related genes that are required for AIEC pathogenesis. Overall the experiments are well designed, and the conclusions are justified by the data provided.

Major points

1) Since there is a *traA* complement experiment in Figure 4, it would be nice to see if the competitive index and fitness advantage in Figure 3b and 3c are reduced when using a *traA/ptraA* complement strain.

Response: This was technically very challenging in our long-term *in vivo* experiments because of the lack of selection for the complementation plasmid. We also encountered off-target genetic scars in the *tra* operon when we tried complementing *traA* *in cis*; thus, this experiment was not feasible. Given that we already demonstrated functional complementation of the knockout, we can safely rule out any unlinked mutations as the cause of the deletion phenotype, validating our conclusions.

2) Figure 4f lacks names of the different *tra* genes referenced in lines 286-287.

Response: We thank the reviewer for noting this omission, which has been fixed.

3) Lines 340-342: Does the higher abundance of *tra* homologs in patient indicate enrichment of *tra* encoded AIEC's in CD patient samples or just enrichment of AIEC's in CD patient samples? The way the data is presented does not separate between these two interpretations. It is unclear how "cell" in gene copies per cell is defined.

Response: We apologize for the lack of clarity on this point. The higher abundance of *tra* homologs indicates an enrichment of *tra* homologs in CD patients. Given that AIEC are known to be enriched in CD patients, it is possible that the *tra* homologues we detected are carried by AIEC strains. However, the lack of clear genetic markers for AIEC makes it challenging to directly associate the *tra* homologues we detected to this pathovar. To clarify this issue, we added the following to the Discussion: "Additionally, we detected an enrichment of *tra* homologues in the microbiomes of CD patients. While it is possible that these homologues are carried by AIEC strains, the power of our *in silico* approach to make this determination is limited by the lack of genetic markers for the AIEC pathogroup."

Reviewer #3 (Remarks to the Author):

Elhenawy et al, aim to determine the mechanism by which adherent-invasive E. coli (AIEC) act as an active disease modifier in Crohn's disease (CD). The authors use Transposon-insertion sequencing (TIS) to conduct an analysis of gene essentiality and/or dispensability in the colonization of the murine gut (cecum). They provide examples of functions they know to be essential/important in vivo. They use RNA-Seq to explore AIEC regulatory changes in vivo by comparing it to an in vitro expression profile. Among the upregulated genes they provide examples of those involved in various forms of metabolism. By comparing TIS and RNA-Seq data they aim to further/again determine genes that mediate functions critical for AIEC pathogenesis and/or colonization. In the 2nd part of the paper the authors focus on a type IV secretion (T4SS) system that was found to be important for survival in vivo. The role of T4SS in biofilm and microcolony formation is studied and the prevalence of T4SS/tra-genes is associated with CD. Overall the study is interesting and has some neat data, however the first half of the study is problematic due to the lack of statistics supported analyses (of either TIS or RNA-Seq data) that have a clear idea/goal in mind. Additionally, the authors only provide some of the data and in a format that makes it inaccessible for anyone that wants to redo or build on the analyses that were done, or simply explore the data that were generated. Moreover, there is little coherence between the first and second part of the study; i.e. there is no clear rationale/analyses that leads from the first to the 2nd part (see for instance point 17 below). All in all, I think the work has some real potential, but right now the two parts feel like they are each a part of a different paper, giving the overall idea that the work is half finished.

Major concerns:

1. The authors describe that "...347 genetic loci [were identified] that enabled AIEC colonization of the host yet were completely dispensable in vitro (Figure 1C)." It is unclear what comparison is made here, e.g. how was it determined that these 347 loci were completely dispensable in vitro? If compared to in vitro data, then what was the comparison exactly? And if it was compared to in vitro was the in vitro sample also re-plated and exposed to gentamycin and chloramphenicol like the in vivo selected libraries? If not than how is this difference in treatment evaluated to affect the in vivo selected library? It is also impossible to assess the amount of variance in the data and the bottlenecks that likely occurred in the experiments. In that sense 3 mice also sounds like a very small number of mice. How stable/confident are the results really (also see point 6)?

Response: The reviewer raises important questions regarding statistical analyses of the TIS data. In the revised manuscript, we have clarified our approach. While TIS statistics were presented in Table S1 of our initial submission, we have now updated this table to include more information as requested by the

reviewer. We used peer-reviewed and published analytic pipelines (Pritchard *et al.* PLoS Genetics, 2014) that have subsequently been implemented in other publications in *PNAS*, *PLoS Pathogens*, and *mBio* so their rigor should not be in question. *In vitro* gene essentiality is based on the frequency of Tn insertions in different genes. The Tn system used in our study targets the genomic ‘TA’ sites. The analysis pipeline previously published by Pritchard JR *et al.* 2014 attempts to distinguish between essential and non-essential genes based on the percentage of ‘TA’ sites disrupted per gene, which typically follows a bimodal distribution that discriminates between ‘essential’ genes (low frequency of Tn insertions) vs ‘non-essential’ genes *in vitro* (high frequency of Tn insertions). See figure below:

The essential genes will have a low frequency of Tn insertions in all libraries (*in vivo* and *in vitro*). The key insight derived by comparing the *in vivo* and *in vitro* libraries arises from the genes that are non-essential *in vitro*, yet conditionally depleted and hence important for fitness in the host due to selection. Both *in vivo* and *in vitro* libraries are selected in the presence of gentamicin and chloramphenicol to minimize the influence of non-host factors on mutant abundance.

As the reviewer correctly indicates, infection bottlenecks are a primary source of spurious results due to genotype loss by drift. However, the pipeline used in our study was previously shown by Pritchard JR *et al.* 2014 to improve the stringency of the TIS analysis, limiting spurious results associated with infection bottlenecks. Although the stochastic loss of genotypes during infection can lead to variation in complexity across libraries, the pipeline we implemented uses computer simulations to mimic the effects of genetic drift on the *in vivo* datasets. Specifically, random sampling is used to generate control datasets that have comparable complexity to the *in vivo* datasets, reducing experimental noise and improving the rigor of the analysis. As reported previously by Pritchard JR *et al.* 2014, a Mann-Whitney test is conducted between the simulated control datasets and the *in vivo* libraries, which generates the P values reported in Table S1. As requested by the reviewer, we now also include additional information in Table S1 regarding the number of reads mapped to each gene and the fraction of TA sites disrupted. As we discussed in the manuscript, some of our hits were reported by other groups to promote bacterial fitness, supporting the findings from our screen. Additionally, we provide correlation plots between our *in vivo* libraries, showing good concordance between the different biological replicates. These data have been provided for both TIS and transcriptomic libraries in Figures S2 and S3.

2. All TIS data should be provided. Including raw read numbers, ratios of input/output, numbers of insertions for each insertion in each genetic region. Without these data it is impossible to recreate and/or assess any of the analyses. These data should be configured in easily accessible files and formats. For instance, a single excel file or csv file. These files should at the least contain each insertion for all genes in the genome, their annotation and gene categories (also see below). Without it, the data are not accessible/usable for anyone else. One of the points of publishing a study like this is that other people can assess and use the data for their own experiments/ideas. This is impossible in the current format.

Response: The reviewer raises an important point. We have deposited the TIS raw data in the SRA database (PRJNA684126) and Featurecounts files of the RNA-sequencing experiment were submitted to GEO (GSM4969708) together with the raw sequencing files. Since all analytic pipelines used are published and are publicly available, there would be no obstacles in rerunning the analysis should one chose to do so. Also, we provide additional information in Table S1 regarding the number of reads mapped per gene, the fraction of TA sites disrupted, and gene annotation of the hits.

3. Transposon construction: since the authors are presenting a new ‘tool’, it would be useful to provide a detailed annotated plasmid map, instead of only a schematic.

Response: A plasmid map is now available in Figure S1.

4. Related to comments 1 and 2, since only a very small part of the data/analysis is provided it’s impossible for anyone to re-run, re-analyze or appropriately interpret the data for instance in Table S1, which makes it hard to determine what the results mean.

Response: Please see response to comment #2

5. Fig 1D lists functional assignments. I can’t find those functional assignments associated with genes COGs, gene names and data anywhere. Thereby one can’t assess and/or rerun the analysis, adjust it or use the data for one’s own study.

Response: As indicted in the manuscript, we ran the COG analysis on the genes that our TIS experiments indicated were important for AIEC fitness *in vivo*. We provided a separate file (Table S2) with the *in vivo* fitness genes that were assigned to COG families by EggNOG. The Table also includes statistics reported by EggNOG.

6. P5-6L110-122: The authors seem to do a lot of cherry picking, i.e. highlighting a bunch of genes that have low abundance, which confirms genes that are known to be important under these or similar conditions. While such highlighting is fine and important. It is only a first step and it certainly does not validate the genomic approach. Proper validation would include a significant number of genotype-phenotype experimental validations in the mouse gut, only than we can assess the real strength and confidence of the data/approach/analysis.

Response: We thank the reviewer for this comment. Following up on a large number of hits in mice would further corroborate our findings, and would have been one way to structure our study. Instead, we decided to focus on the T4SS hit because its expression was upregulated *in vivo*. Furthermore, we hypothesized that this hit might provide critical new insight into AIEC colonization outside of the metabolic requirements for colonization that have been studied previously. In our view, this approach proved successful and provides some validation to the combined TIS/RNA-seq findings. Also, we highlighted in the manuscript many examples of genes identified in our screen that were found to be important for gut colonization by other groups, which validates our approach and provides further support to our findings.

7. A STRING network is presented in FigS2. It is unclear how this network was extracted from the database as the required information to do this is not presented, i.e. I couldn't find it. It is thereby unclear how the STRING network was generated. Importantly what is the value of the STRING network? There is no real network-analysis here (P27L512-514: This does not describe/define a network analysis). The network could be valuable just for data organization, and while it looks cool its value here is not clear.

Response: The purpose of this analysis was to show the functional links between the *in vivo* fitness genes detected in our high-throughput assay. We used the STRING database and ran the analysis with the default settings. Previous Figure S2 was generated and exported from STRING without manipulation. However, we acknowledge that the Figure adds limited value and have removed it from the revised manuscript. The *in vivo* fitness genes presented in Table S1 remain available for the scientific community if there is interest in running such analysis in the future.

8. P7L125-127. "The COG analysis...". What COG analysis is performed here? If a proper COG analysis is performed, what then are the statistics or enrichment scores? There could definitely be enrichment here, and if you write analysis then do the statistics. If not then say there's a trend or a tendency for something. I'm fine with the authors just cherry picking some data here that helps support some ideas, but than just say that that's what you're doing. But preferably do some proper analysis, like an enrichment analysis, which can have its own shortcomings, but there are several different options to do some good analyses for pattern discovery.

Response: We thank the reviewer for this comment and suggestion. To perform the COG analysis, we used the EggNOG database for functional prediction. The raw data for this analysis is now provided in Table S2 (for TIS hits) and Table S4 (for RNA-seq hits), including the e-values and scores reported by EggNOG. This is different from the enrichment analysis, which queries the dataset to identify the specific pathways that are significantly enriched in the dataset. To conduct a pathway enrichment analysis using published tools (like GSEA), we will need a gmt file specific for our AIEC strain, which is currently unavailable. However, our data will be publicly available for members of the scientific community should they desire to run a pathway enrichment analysis in the future.

9. P7-8L125-150. Again, a lot of what the authors write about seems to be cherrypicked from the data, while there don't seem to be any ideas or hypotheses that are generated from any specific analysis. Also there doesn't seem to be any unifying and/or new theme that emerges from these descriptions. The presented 'analysis' thereby doesn't add any value on top of the generated data. It is evident that things are/will be different *in vivo*, however the 'analysis' does not create much order in the data that gets presented.

Response: We politely disagree with the reviewer's interpretation that nothing new emerges from our work. Our goal was not to show that bacterial fitness and transcriptomes differ between *in vitro* and *in vivo* conditions. Of course, this would be true. Our goal was to identify *in vivo*-conditionally essential genes for a disease-associated bacterium whose infection biology remains poorly characterized. Previous work by us and others highlight the important role of metabolism in the ability of AIEC to adapt to the gut environment and outcompete commensals. Our work presented in the current manuscript is the first comprehensive analysis of the metabolic pathways required by AIEC to colonize the gut, the first comprehensive genome-scale TIS experiments for AIEC in a host, and the first to explore the role of type 4 secretion and biofilm formation *in vivo* as a major colonization determinant. While we chose to focus on identifying the role of type 4 secretion system in AIEC virulence, our publicly available datasets will offer the scientific community more opportunities to conduct further analyses, and better understand the infection biology of this important group of pathogens.

10. P8L167 “Our TIS screen revealed the presence of several two-component systems that are required for AIEC fitness *in vivo*, suggesting that AIEC undergo regulatory adaptation to the host environment.” Indeed, I don’t think anyone will deny things change *in vivo*. But this feels like an example of Chekov’s gun : ‘ If in the first act you have hung a pistol on the wall, then in the following one it should be fired’. However, nowhere in the proposal do these two component systems (TCS) return to fulfill a role. So why so much emphasis on these TCSs?

Response: We disagree that we have placed untoward emphasis on TCS in the manuscript. We name the TCS important for *in vivo* colonization as identified in our data in only two lines of the manuscript. We also believe that this knowledge will be valuable for future studies in this field, consistent with similar reports of TCS being conditionally essential for several other pathogens during infection.

11. P8L170: “Three days after AIEC infection, total RNA was isolated from the ceca of mice and sequenced to characterize AIEC gene expression (Figure 2A).” What was the quality of the RNA isolated from these *in vivo* samples? RNA-quality from *in vivo* samples can be wildly different and of bad quality, affecting any downstream analysis. How does the quality relate to the *in vitro* quality? How was this assessed? How did Streptomycin treatment affect transcription? What was the *in vitro* condition to generate *in vivo* DE genes? Was there an *in vitro* Spectinomycin control to assess the effects of Spectinomycin? What was the growth phase of the *in vitro* sample and how does growth phase affect the observed *in vivo* DE genes?

Response: All RNA samples were analyzed on an Agilent bioanalyzer system with all RIN values above 7, including the *in vivo* samples. This information is now included in the manuscript. We also show that the different *in vivo* transcriptomes are strongly correlated (Figure S3). We performed the infection 24 h post streptomycin treatment to avoid directly exposing AIEC to streptomycin; however, it is worth noting that our AIEC strain is naturally resistant to streptomycin (as outlined in the methods). As indicated in the methods, we grow AIEC *in vitro*, which is then used to prepare inoculum for infections. A paired culture sample is sub-cultured *in vitro* for approximately the same number of generations. We based these calculations on a previous study by Rang CU *et al.* (1999) that determined the generation time of *E. coli* in streptomycin-treated mice to be ~130 min.

12. Similar to the TIS data above, where are the raw read numbers for the RNA-Seq data, the processed numbers (e.g. featurecounts tables, DEseq files, annotation and COGs?), so people can actually easily use and evaluate it? The data is just not easily accessible and/or unexplorable right now. Overall no effort seems to have gone into making any of these data TIS/RNA-Seq) easily explorable and accessible, which goes against the FAIR principles and which is a shame.

Response: We thank the reviewer for urging us to make our data as accessible as possible. We have now included the information the reviewer was requesting. Table S1 for the TIS data includes the number of reads mapped per gene, and the fraction of “TA” sites disrupted. For the RNA-sequencing data, Featurecounts tables were submitted to GEO (GSM4969708) together with the raw sequencing files. EggNOG data for the COG analysis has been provided as Tables S2 and S4 for TIS and RNA-sequencing hits, respectively.

13. I couldn’t find the data the venn diagram is based on. Which genes are in the overlaps and non-overlaps? What are their functions, COGs, values, how do these data/types relate to each other?

Response: The common genes between the two datasets were originally supplied in Table S3. However, we decided to decrease the emphasis on the overlap between the two datasets (TIS, RNA-seq) as the

reviewer suggested and focus only on the upregulated *in vivo* fitness genes (instead of upregulated and downregulated genes), now provided in Table S5. Therefore, the Venn diagram was removed in the revised manuscript.

14. Overall Figures 1 and 2 are largely uninformative. There is not much one would take away from it.

Response: We respectfully disagree with the reviewer's position. Figures 1 and 2 represent our comprehensive genomic analysis from both *in vivo* Tn-seq and RNA sequencing experiments. Indeed, it is these data that led us to identify the role of T4SS in AIEC pathogenesis. Additionally, these data will be important for the members of the scientific community who have interest in uncovering new mechanisms of AIEC virulence.

15. P10L187 *"Among these, 1,634 genes could be assigned to 17 COGs, with nearly half (46%) belonging to metabolic pathways, consistent with the TIS data that revealed the importance of metabolic adaptability."* Again, this would require a proper statistical/enrichment analysis to show that this suggestion is actually supported. For instance, if a large proportion of your genome is associated with metabolism, then a large proportion of DE genes could naturally be associated with metabolism. And again, no reader would be able to do this analysis either, because the proper files/info are missing.

Response: See response to comment 8.

16. P10L192 – 218 *feels like cherry picking. No real computational/network analysis here, with no validation, and no way to easily find these results in the data. The authors suggest interesting things, but up to now it's all observations, that do not contribute to a coherent story.*

Response: We are not reporting a network analysis in this section, rather we are discussing the results of the DE analysis. Given the important role of metabolism in AIEC virulence, we are highlighting the pathways involved in consuming gut metabolites. Importantly, we compared our results to the findings of other groups and found common metabolic signatures, which supports our approach and appropriately recognizes foundational knowledge.

17. P11L223. *"To uncover new genes and pathways that contribute to AIEC infection biology, we cross-referenced the list of genes that were differentially regulated in the gut to the set of genes that were required for in vivo growth identified by TIS. Our rationale was that genes essential for gut colonization and also differentially upregulated upon entering the gut environment likely mediate critical functions in AIEC pathogenesis and colonization."*

This is not a rationale that holds water. It is unclear why a TIS gene (i.e. one with a fitness defect) that is also differentially regulated would be more important than a non-differentially regulated TIS gene. There are plenty of studies that show a clear disconnect between genes important for fitness, and those that are transcriptionally regulated. There are also several examples of analyses that can combine these data types and then help extract important patterns from the data. The problem with 'the rationale' is that it suggests that the authors used some statistical/network analyses to get to the next and most important part of the paper, which focuses on the T4SS. In reality the presented analyses in the 1st part does not lead to the 2nd part of the paper. The most important problem here is that there is no clear reasoning why the T4SS is picked out of the large number of TIS genes and suddenly becomes the focus of the data-set and paper. There is nothing wrong with focusing on the T4SS, I think it's a very interesting phenotype that gets presented in the remaining part of the paper, however the presented TIS and RNA-Seq data in the first half of the paper overall have very little, to nothing to do with the second part of the paper. The authors could basically have started the paper with "T4SS was identified in a TIS screen as important for survival in the cecum." And then only present the 2nd half of the paper. Right now, in the way the 1st half

of the paper is analyzed and presented, I do not see the added value to the manuscript.

Response: The reviewer raises a valid point; there are certainly examples of differences in genes required for fitness and DE genes. However, as we explained in the manuscript, the goal of the combined ‘omic approaches was to prioritize studying the genes that are identified by several assays as likely key players in bacterial virulence. Combined ‘omics has been successfully used in this manner by different groups (Serafini A *et al* 2019, Vogt SL *et al* 2019, Chatterjee A *et al* 2020). In this study, our TIS assay identified the genes important for *in vivo* fitness. However, we were interested in the genes that are also upregulated shortly post infection because these genes are likely playing key roles in the early establishment of AIEC colonization. Such genes included T4SS genes, which led us to investigate the mechanistic role of this secretion system in AIEC fitness. We decided to focus on T4SS because the role of this machinery in AIEC virulence was not investigated before, to the best of our knowledge. We believe there is great value in reporting these datasets so that the field may build on and advance these findings expeditiously.

18. P12-Table S3 –*This table does not contain clearly presented cross-referenced- TIS/RNA-Seq data.*

Response: The original Table S3 listed the genetic hits that were common between the two datasets. In the revision, we focus only on the upregulated genes, which are presented in new Table S5.

19. P12L228. *“This analysis identified 149 genes that met these criteria, with the majority of them controlling metabolic functions, confirming the pivotal role of metabolic adaptation in promoting AIEC in vivo fitness.”– Again I don’t think it does, you would need to do the proper analysis to show its different from the distribution you’d expect based on the genome distribution.*

Response: We thank the reviewer for this comment. We toned down this section and highlighted only a few examples of metabolic genes that were common between the datasets.

20. P11L229 – 235 – *Subsequently, as above in points 9 and 16 these are more ad hoc explanations for some of the data, which does not equal proper analysis*

Response: It’s unclear if the reviewer is criticizing the peer-reviewed pipelines used for TIS and RNA-seq analyses or suggesting additional analyses (i.e. pathway enrichment). Pathway enrichment is not something we were claiming in this section, we are simply highlighting genes that were found to be important for AIEC gut colonization based on our experimental design. However, our datasets will be available for the scientific community to run more informative analyses including pathway enrichment and metabolic network modeling.

21. P13L265 and P17L314. *It is unclear whether microcolonies are the same or similar to biofilms. The importance of microcolonies in the infection is also not clear; they may be better at resisting antibiotic mediated clearance, or maybe they provide an advantage during invasion of resident populations. This is a missing link in the paper; i.e. what is the true advantage of the phenotype? On P19L342. The authors state: “Together, these data suggest that the T4SS promotes the pathogenesis of CD-associated AIEC by facilitating their long-term persistence in biofilms in the host setting.” Also here biofilms may not be the right wording as there is mostly evidence for the importance of microcolony formation.*

Response: The reviewer raises a valid point. It is difficult to disentangle the phenotypic significance of microcolonies and biofilms. In the revised manuscript, we note that our findings implicate microcolonies and/or biofilms. The *traA* mutant was less persistent in the gut and simultaneously less efficient at forming microcolonies *in vivo*. We believe, given the complexities of the CD gut environment, that these

structural features may confer different benefits under different conditions. We did not use antibiotics post-colonization in our experiments, so this comment is not relevant to our experimental design. Biofilms are bacteria embedded in an exopolysaccharide-rich matrix. We showed that AIEC microcolonies are abundant in cellulose, which is a major component of *E. coli* biofilms (see Figure 4B). Moreover, we show that cellulose formation is required for microcolony formation. Thus, in this context, AIEC microcolonies meet the criteria used for identifying bacterial biofilms. We agree with the reviewer in that identifying the ‘true advantage of the phenotype’ has not been achieved; however, our findings motivate future studies to unravel this intriguing question. We note that determining the fitness advantage of microcolony formation *in vivo* has been a long-standing challenge for many enteric pathogens: for example, in *V. cholerae*, TCP mediates microcolony formation *in vivo* and promotes colonization, but the mechanism explaining how microcolony formation facilitates colonization has not been revealed despite 30+ years of research efforts.

22. P19Fig6B. This analysis is not clear and no (raw) data is provided to redo/rework/assess the analysis; e.g which genomes were selected, what is the associated metadata? While the finding/association between tra-genes and CD is interesting, it is not possible to assess whether the comparison/association that was made is fair.

Response: This analysis was conducted using the previously published metagenomic tool, MetaQuery. This application interrogates ~2000 publicly available human metagenomes for the enrichment of the queried genes. As suggested by the reviewer, in the revised manuscript we provided more details on the analysis and the metagenomes interrogated.

REVIEWERS' COMMENTS

Reviewer #1 (Remarks to the Author):

This is a very well performed study. I have no further concerns.

Reviewer #3 (Remarks to the Author):

The manner in which the manuscript is presented has improved considerably. Importantly, this manuscript contains some really nice datasets which will be useful for the community/field. However, I still find that the data has some important, easy to solve, accessibility issues which I detailed below. In addition I have a couple of small other remarks.

1.

It would be really useful for the community if the investigators create 1 Supplemental Table that combines all the relevant information into a single worksheet. Right now I have to navigate between 5 Supplemental files and even more worksheets to get an idea of whether a gene is significant *in vivo*, what its gene name is, its annotation, what its COG category is and whether it is associated with a transcriptional change *in vivo* or not. It will probably take the authors less than an hour to combine these data into a single worksheet and thereby provide an immensely great service to the community/field. I also recommend you add all the DE info and not just the up regulated data.

2.

There are multiple gene identifiers used throughout the paper that I was unable to locate in the supplemental datasheets. This includes: *fepD*, *fhuE*; *frdA*, B, C, D; *NorR*,V,W. Again this could be solved with a single datasheet that lists the genenames into a single searchable column and besides all the other info (e.g. TIS data, DE expression data et cetera).

3. P5 L112 (Numbers refer to the marked-up file).

I am still unsure when a gene is called significant *in vivo*. In the Material and methods section the authors write: "Mutants showing less than 0.5-fold reduction in abundance *in vivo* compared to the *in vitro* control ($P < 0.01$) were considered to be significantly attenuated in colonizing the host."

Does this mean a gene needs to be significant in one mouse, two mice or three mice before it is called significant? It would be useful if Table S1 has a column that indicates which genes are considered significant by the authors. Right now it lists p-values/mouse but again it's unclear when something is considered significant.

4. P7 L154 / P11L245 (Numbers refer to the marked-up file).

Listed percentages like 50% (P7 L154) (as well as other related numbers in the TIS and DE sections) as well as those in Figure 1D and 2C should be put into perspective of the distribution of these categories in the genome. For instance 50% metabolic genes significant in TIS data indeed sounds high, but it only makes sense if the reader understands what the distribution of metabolic genes and other COG categories are across the genome. If the genome overall has 50% metabolic genes, it's not higher than expected to get 50% significant TIS hits, if the genome consists of 30% metabolic genes 50% indeed sounds higher, which you can easily statistically test for. Potentially, the authors can add the genome distribution of COGs + significance to the figures, which right away can indicate to the reader what is more or less important.

5. P8 L170 (Numbers refer to the marked-up file).

than = that

6. P18 L427 (Numbers refer to the marked-up file).

"was isolated from a the gut of a CD patient" : remove first "a"

Reviewer #1 (Remarks to the Author):

This is a very well performed study. I have no further concerns.

Response: We thank the reviewer for their positive comments.

Reviewer #3 (Remarks to the Author):

The manner in which the manuscript is presented has improved considerably. Importantly, this manuscript contains some really nice datasets which will be useful for the community/field. However, I still find that the data has some important, easy to solve, accessibility issues which I detailed below. In addition I have a couple of small other remarks.

1. It would be really useful for the community if the investigators create 1 Supplemental Table that combines all the relevant information into a single worksheet. Right now I have to navigate between 5 Supplemental files and even more worksheets to get an idea of whether a gene is significant *in vivo*, what its gene name is, its annotation, what its COG category is and whether it is associated with a transcriptional change *in vivo* or not. It will probably take the authors less than an hour to combine these data into a single worksheet and thereby provide an immensely great service to the community/field. I also recommend you add all the DE info and not just the up regulated data.

Response: We thank the reviewer for their positive comments and suggestions that have ultimately improved the manuscript. As requested, we have collated all datasets into one, Supplementary Data 1.

2. There are multiple gene identifiers used throughout the paper that I was unable to locate in the supplemental datasheets. This includes: *fepD*, *fhuE*; *frdA*, B, C, D; *NorR*, V, W. Again this could be solved with a single datasheet that lists the genenames into a single searchable column and besides all the other info (e.g. TIS data, DE expression data et cetera).

Response: We included gene names /annotations in Supplementary Data 1.

3. P5 L112 (Numbers refer to the marked-up file).

I am still unsure when a gene is called significant *in vivo*. In the Material and methods section the authors write: "Mutants showing less than 0.5-fold reduction in abundance *in vivo* compared to the *in vitro* control ($P < 0.01$) were considered to be significantly attenuated in colonizing the host."

Does this mean a gene needs to be significant in one mouse, two mice or three mice before it is called significant? It would be useful if Table S1 has a column that indicates which genes are considered significant by the authors. Right now it lists p-values/mouse but again it's unclear when something is considered significant.

Response: By our definition, hits have to be significant in each independent biological replicate/animal to be considered required for *in vivo* fitness. We added a column in Supplementary Data 1 showing for each locus whether it is required for fitness and/or differentially-regulated in the gut.

4. P7 L154 / P11L245 (Numbers refer to the marked-up file).

Listed percentages like 50% (P7 L154) (as well as other related numbers in the TIS and DE sections) as well as those in Figure 1D and 2C should be put into perspective of the distribution of these categories in the genome. For instance 50% metabolic genes significant in TIS data indeed sounds high, but it only makes sense if the reader understands what the distribution of metabolic genes and other COG categories are across the genome. If the genome overall has 50% metabolic genes, it's not higher than expected to get 50% significant TIS hits, if the genome consists of 30% metabolic genes 50% indeed sounds higher, which you can easily statistically test for. Potentially, the authors can add the genome distribution of COGs + significance to the figures, which right away can indicate to the reader what is more or less important.

Response: We thank the reviewer for their suggestion. We conducted a genome-wide COG analysis and used that to compare the distribution of genes required for fitness and the differentially regulated loci across the COG groups (included in Figure 2). We found a higher representation of metabolism-related COG families in our datasets compared to the genome. We highlighted those examples in the manuscript.

5. P8 L170 (Numbers refer to the marked-up file).

than = that

Corrected.

6. P18 L427 (Numbers refer to the marked-up file).

"was isolated from a the gut of a CD patient" : remove first "a"

Corrected.